# Synaptic mechanisms of top-down control in the non-lemniscal inferior colliculus

**Hannah M Oberle[1,2], Alexander N Ford[1], Deepak Dileepkumar[1], Jordyn Czarny[1], Pierre F Apostolides[1,3]***

[1]Kresge Hearing Research Institute & Department of Otolaryngology, University of Michigan, Ann Arbor, United States; [2]Neuroscience Graduate Program, University of Michigan, Ann Arbor, United States; [3]Molecular and Integrative Physiology, University of Michigan Medical School, Ann Arbor, United States

**Abstract** Corticofugal projections to evolutionarily ancient, subcortical structures are ubiquitous across mammalian sensory systems. These 'descending' pathways enable the neocortex to control ascending sensory representations in a predictive or feedback manner, but the underlying cellular mechanisms are poorly understood. Here, we combine optogenetic approaches with in vivo and in vitro patch-clamp electrophysiology to study the projection from mouse auditory cortex to the inferior colliculus (IC), a major descending auditory pathway that controls IC neuron feature selectivity, plasticity, and auditory perceptual learning. Although individual auditory cortico-collicular synapses were generally weak, IC neurons often integrated inputs from multiple corticofugal axons that generated reliable, tonic depolarizations even during prolonged presynaptic activity. Latency measurements in vivo showed that descending signals reach the IC within 30 ms of sound onset, which in IC neurons corresponded to the peak of synaptic depolarizations evoked by short sounds. Activating ascending and descending pathways at latencies expected in vivo caused a NMDA receptor-dependent, supralinear excitatory postsynaptic potential summation, indicating that descending signals can nonlinearly amplify IC neurons' moment-to-moment acoustic responses. Our results shed light upon the synaptic bases of descending sensory control and imply that heterosynaptic cooperativity contributes to the auditory cortico-collicular pathway's role in plasticity and perceptual learning.

**\*For correspondence:**
piaposto@med.umich.edu

**Competing interest:** The authors declare that no competing interests exist.

## Editor's evaluation

This study provides evidence for NMDA-dependent nonlinear integration of corticofugal inputs with feedforward sound-driven inputs in inferior colliculus. This unexpected property will be important for understanding the role of cortical feedback in sound processing.

## Introduction

The auditory system is organized as a network of feedback loops, such that most central auditory nuclei receive descending projections from higher levels of the processing hierarchy (*Diamond et al., 1969*; *Saldaña et al., 1996*; *Winer et al., 1998*; *Winer et al., 2001*; *Doucet et al., 2003*; *Coomes and Schofield, 2004*; *Schofield et al., 2006*; *Suthakar and Ryugo, 2017*). The auditory cortex is a major source of excitatory (glutamatergic) descending projections, with the density of descending fibers often rivaling that of ascending fiber tracts (*Winer et al., 2001*; *Winer, 2006*; *Stebbings et al., 2014*). These corticofugal projections likely play a major role in hearing by providing an anatomical substrate for 'top-down' information to control early acoustic processing. Indeed, stimulating or silencing the auditory cortex in vivo changes spontaneous and sound-evoked activity throughout the

central auditory system (*Massopust and Ordy, 1962*; *Ryugo and Weinberger, 1976*; *Yan and Suga, 1998*; *Yan and Suga, 1999*; *Nwabueze-Ogbo et al., 2002*; *Xiao and Suga, 2002*; *Yu et al., 2004*; *Nakamoto et al., 2008*; *Nakamoto et al., 2010*; *Anderson and Malmierca, 2013*; *Kong et al., 2014*; *Vila et al., 2019*; *Blackwell et al., 2020*; *Qi et al., 2020*), indicating that high-level activity regulates the moment-to-moment function of subcortical auditory circuits. However, little is known regarding the biophysical properties of auditory corticofugal synapses, nor do we understand how descending signals are integrated with ascending information. Given that synaptic dynamics and pathway integration are fundamental building blocks of neural circuit computations (*Abbott et al., 1997*; *Zucker and Regehr, 2002*; *Stuart and Spruston, 2015*), addressing these knowledge gaps is necessary to understand how the auditory cortex exerts control over early auditory processing.

Of particular interest is the descending projection from auditory cortex to the inferior colliculus (IC), a midbrain hub important for sound localization, speech perception, and an early site of divergence for primary and higher-order auditory pathways (*Masterton et al., 1968*; *Krishna and Semple, 2000*; *Champoux et al., 2007*; *Sinex and Li, 2007*; *Joswig et al., 2015*). The IC is generally subdivided into a 'lemniscal' central core and 'non-lemniscal' dorsal and lateral shell regions whose neurons have distinct afferent and efferent connections (*Faye-Lund and Osen, 1985*; *Loftus et al., 2008*; *Ayala et al., 2015*; *Chen et al., 2018*). Whereas central IC neurons project mainly to the primary auditory thalamus (ventral medial geniculate nucleus; *Mellott et al., 2014*; *Oliver, 1984*), shell IC neurons preferentially project to secondary, higher-order auditory thalamic nuclei that subsequently funnel acoustic information to the amygdala and striatum (*Oliver and Hall, 1978*; *LeDoux et al., 1990*; *Bordi and LeDoux, 1994*; *Mellott et al., 2014*; *Cai et al., 2019*; *Ponvert and Jaramillo, 2019*). Auditory cortico-collicular axons terminate primarily in the shell IC, with comparatively fewer fibers in the central IC (*Bajo et al., 2019*; *Chen et al., 2018*; *Lesicko et al., 2016*; *Song et al., 2018*; *Winer et al., 1998*; *Xiong et al., 2015*; but see *Saldaña et al., 1996*). Thus, auditory cortico-collicular synapses seem uniquely positioned to modulate acoustic signals destined for limbic circuits supporting learned valence and habit formation; this prediction is further supported by the fact that chemical ablation of auditory cortico-collicular neurons selectively impairs certain forms of auditory perceptual learning while sparing the performance of previously learned task associations (*Bajo et al., 2010*). Nevertheless, little is known regarding how auditory cortico-collicular synapses control activity in single IC neurons. Intriguingly, auditory cortex inactivation typically does not abolish IC neuron sound responses, but rather causes divisive, non-monotonic changes in receptive field properties and feature selectivity (*Yan and Suga, 1999*; *Nakamoto et al., 2008*; *Nakamoto et al., 2010*; *Anderson and Malmierca, 2013*). Thus, descending transmission might operate in part via heterosynaptic interactions, perhaps by controlling how IC neurons respond to ascending acoustic inputs.

Here, we employ electrophysiology and optogenetic approaches to identify how auditory cortico-collicular synapses transmit descending signals, and to understand how descending synapses control IC neuron responses to ascending inputs. We find that the majority of shell IC neurons receive monosynaptic inputs from auditory cortex and often integrate information from multiple distinct corticofugal axons. Synaptic latency measurements in vivo show that descending excitation reaches IC neurons ~5–7 ms after spike initiation in auditory cortex, such that cortical feedback will rapidly follow the onset of acoustically driven excitation. Somewhat surprisingly, NMDA receptors only modestly contribute to descending transmission. By contrast, excitatory intra-collicular synapses from the central IC, which are probably a major source of ascending acoustic signals to shell IC neurons, had a much larger NMDA component. Consequently, appropriately timed activity in ascending and descending pathways integrates supralinearly owing to the cooperative activation of NMDA receptors. Our data reveal a key role for heterosynaptic nonlinearities in the descending modulation of early acoustic processing. In addition, the results place important biophysical constraints on the synaptic learning rules that might support the auditory cortico-collicular pathway's role in experience-dependent plasticity and perceptual learning.

## Results

### Auditory cortico-collicular synapses robustly target superficial IC neurons

Corticofugal axons are predominantly restricted to the shell IC, but little is known about the extent of functional synaptic connectivity between auditory cortex and IC neurons. We tested how auditory cortico-collicular synapses impact IC neurons by transducing the optogenetic activator Chronos in the auditory cortex of mice via intracranial AAV injections (*Figure 1A*), and 2–4 weeks later, performing in vivo whole-cell recordings from the ipsilateral IC of urethane anesthetized mice. GFP-tagged, Chronos-expressing auditory cortico-collicular axons were primarily (though not exclusively) restricted to the ipsilateral dorsal-medial and lateral IC (*Figure 1—figure supplement 1*), in agreement with previous studies. Single flashes of blue light from an optic fiber positioned over auditory cortex (1–5 ms duration) reliably triggered excitatory postsynaptic potentials (EPSPs) in n = 21/38 IC neurons recorded from N = 8 mice (*Figure 1B*). EPSPs were primarily observed in superficial IC neurons, with the majority of unresponsive neurons located more ventrally, presumably in the central IC (*Figure 1C*; mean depth of cortico recipient and nonrecipient neurons: 220 ± 23 vs. 354 ± 38 μm from surface, p=0.0036, Kolmogorov–Smirnov test). A qualitatively similar result was obtained by injecting a trans-synaptic cre virus in auditory cortex of Ai14 tdTomato reporter mice (*Zingg et al., 2017*; *Zingg et al., 2020*): The majority of anterograde, trans-synaptically labeled IC neurons were located in the dorso-medial and lateral shell regions (N = 3 mice; *Figure 1—figure supplement 2*), although a few central IC neurons were indeed labeled. Thus, whereas the non-lemniscal shell IC likely receives the most profuse descending inputs from auditory cortex, monosynaptic cortical signals nevertheless likely reach a few neurons in the lemniscal, central IC.

EPSPs in superficial IC neurons had an onset latency of 10.3 ± 0.7 ms following light stimulation (*Figure 1D*), a peak amplitude of 3.5 ± 0.7 mV (*Figure 1E*), and a full-width at half-maximum of 18.4 ± 1.5 ms (*Figure 1F*). Interestingly, EPSP amplitudes varied over two orders of magnitude across different cells and were occasionally large enough to drive IC neurons beyond spike threshold (*Figure 1—figure supplement 3A*). These data indicate that bulk activation of corticofugal neurons triggers potent EPSPs in superficial IC neurons. However, under behaviorally relevant conditions, the extent of synaptic depolarization provided by descending inputs will depend on the rate and synchrony of corticofugal neuron firing.

Because the auditory cortex projects to many subcortical targets besides the IC, in vivo stimulation could drive polysynaptic excitation onto IC neurons that would complicate estimates of monosynaptic connectivity. We thus prepared acute IC brain slices from mice injected with Chronos in auditory cortex to quantify the functional properties of descending synapses in a more controlled setting. We targeted whole-cell current-clamp recordings specifically to neurons in the dorso-medial shell IC as this region shows the highest density of corticofugal axons (*Song et al., 2018*; *Figure 1—figure supplements 1 and 2*). Stimulating auditory cortico-collicular axons via single blue light flashes delivered through the microscope objective (1–10 ms duration) drove EPSPs in n = 78 neurons from N = 40 mice (*Figure 1G and H*). EPSPs had short-latency onsets following photostimulation (*Figure 1I*; 3.0 ± 0.1 ms), indicating a monosynaptic rather than polysynaptic origin. EPSPs in vitro had a similar range and mean peak amplitude as those recorded in vivo (*Figure 1J*; 2.97 ± 0.35 mV, p=0.3, Mann–Whitney test), and similarly could drive spikes in a subset of recordings (*Figure 1—figure supplement 3B*). Although the EPSP half-width was significantly slower in vitro compared to in vivo (*Figure 1K*; 39.6 ± 2.3 ms, p<0.001, rank-sum test), this result is not surprising: The constant barrage of synaptic inputs in vivo is expected to generate a 'high conductance state' that accelerates the membrane time constant (*Destexhe et al., 2003*).

In a separate set of experiments (n = 24 neurons from N = 10 mice), we quantified the kinetics of auditory cortical excitatory postsynaptic currents (EPSCs) as they appear at the soma using voltage-clamp recordings (*Figure 1—figure supplement 4*; peak amplitude = 48.1 ± 10.1 pA; weighted decay time constant = 5.8 ± 0.8 ms; 10–90% rise time of 1.3 ± 0.1 ms). Altogether, our results show that auditory cortico-collicular synapses substantially depolarize shell IC neurons independent of network-level, polysynaptic activity. In addition, EPSP kinetics are such that descending excitation will undergo significant temporal summation at firing rates observed during sound-evoked activity of auditory cortico-collicular neurons (20–50 Hz; *Williamson and Polley, 2019*).

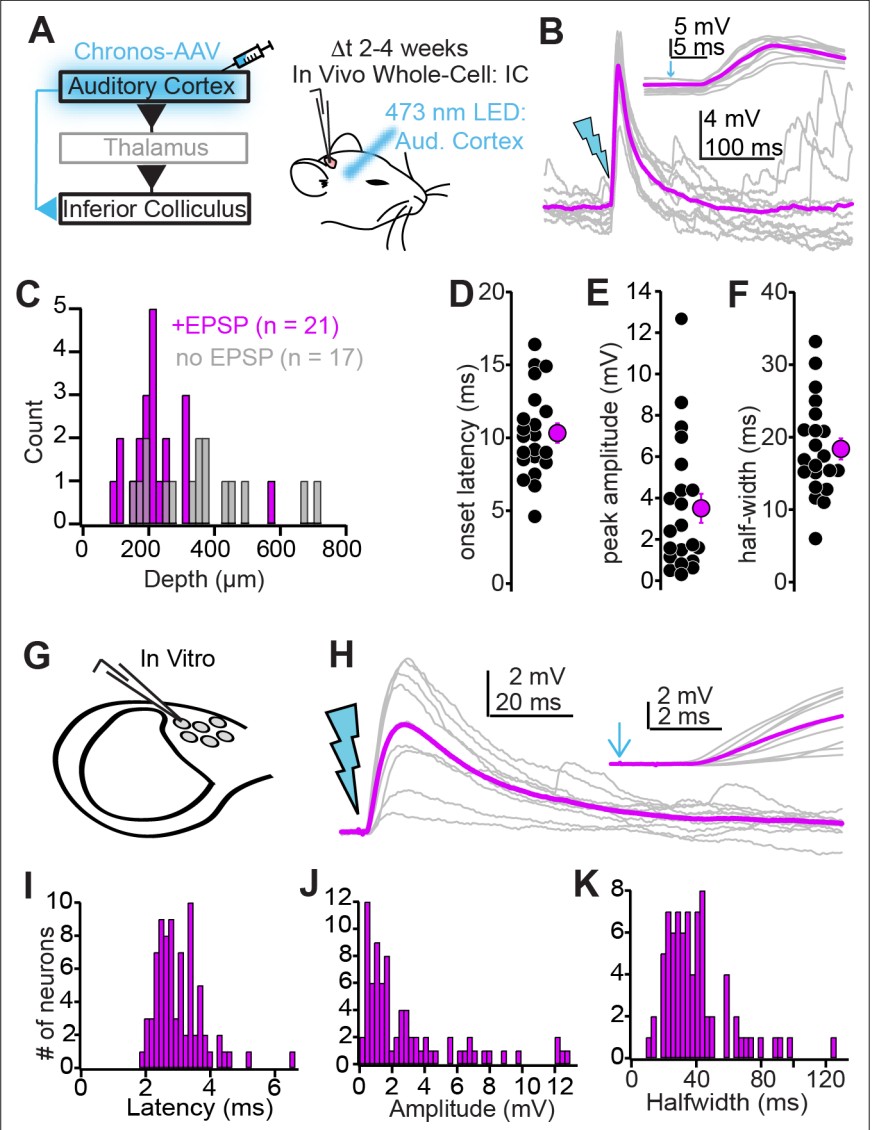

**Figure 1.** Biophysical properties of auditory cortico-collicular synapses. (**A**) Cartoon of experiment. In vivo whole-cell recordings are obtained from inferior colliculus (IC) neurons 2–4 weeks following Chronos injections; an optic fiber is positioned above the auditory cortex. (**B**) Example EPSPs following in vivo optogenetic stimulation. Gray traces are single trials; magenta is average. Inset is the EPSP rising phase at a faster timebase, arrow denotes light onset. (**C**) Dorsal-ventral locations (relative to dura) for IC neurons where auditory cortical stimulation did (magenta) and did not (gray) evoke an EPSP. (**D–F**) Summary of EPSP onset (**D**) amplitude (**E**), and half-width (**F**). (**G**) Whole-cell recordings obtained from dorso-medial shell IC neurons in vitro. (**H**) EPSPs evoked by in vitro optogenetic stimulation (2 ms light flash). Inset: EPSP rising phase. (**I–K**) Histograms of EPSP onset (**I**), amplitudes (**J**), and half-widths (**K**) in vitro.

The online version of this article includes the following figure supplement(s) for figure 1:

**Figure supplement 1.** Auditory cortical axons are preferentially distributed in the non-lemniscal shell inferior colliculus (IC).

**Figure supplement 2.** Anterograde trans-synaptic labeling of auditory cortical targets in the inferior colliculus (IC).

**Figure supplement 3.** Auditory cortico-collicular EPSPs can trigger spikes in inferior colliculus (IC) neurons in vivo and in vitro.

**Figure supplement 4.** Auditory cortico-collicular transmission in voltage-clamp.

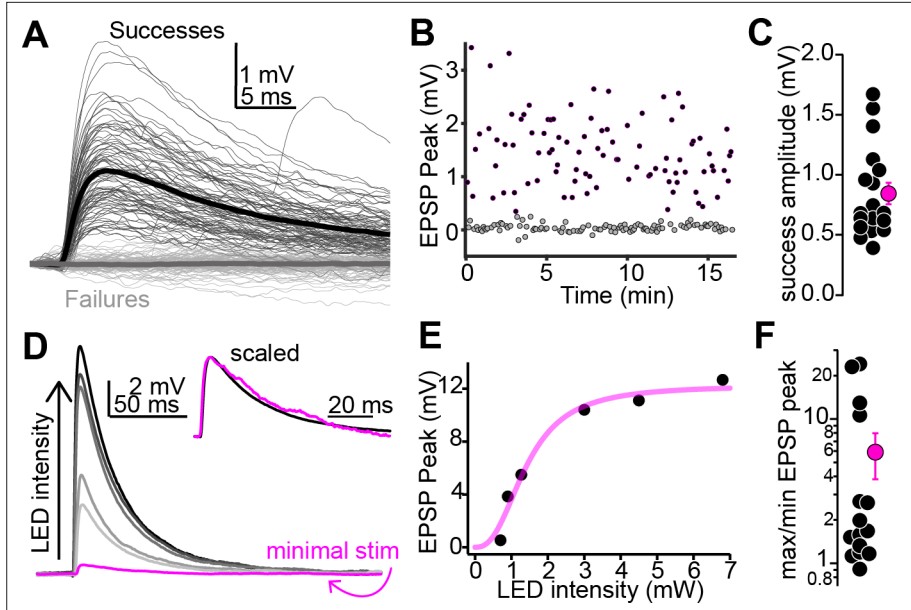

**Figure 2.** Multiple auditory cortico-collicular fibers converge onto single-shell inferior colliculus (IC) neurons. (**A**) Example threshold optogenetic stimulation experiment. Thin black and gray traces are successes failures, respectively. Thick traces are averages. (**B**) Diary plot from the experiment in panel (**A**). black and gray symbols are successes and failures, respectively. (**C**) Summary of putative unitary EPSP amplitudes. (**D**) Magenta: putative unitary EPSP (average of successes recorded at threshold stimulation). Black and gray traces: averages of successes at increasing LED intensity. Data are from a different neuron as in (**A**). Inset: peak scaled and onset aligned EPSPs recorded at threshold and maximal intensity, revealing an identical time course. (**E**) EPSP peak amplitude is plotted as a function of LED intensity for the recording in (**D**). Magenta line is Hill fit. (**F**) EPSP amplitude ratios at maximal and threshold LED intensities. Most values are >1, indicating convergence of at least two fibers. Of note, data are on a log scale.

The online version of this article includes the following figure supplement(s) for figure 2:

**Figure supplement 1.** Excitatory postsynaptic current (EPSC) rise times and half-widths are similar across minimal and maximal stimulation intensities.

## Synaptic strength reflects presynaptic convergence rather than unitary EPSP amplitudes

EPSP amplitudes spanned two orders of magnitude under our conditions (*Figure 1E and J*). Does this variability reflect a differential potency of individual synapses, or alternatively, differences in the number of presynaptic auditory cortical axons impinging onto individual shell IC neurons? We first estimated unitary EPSP amplitudes using a minimal stimulation paradigm designed to activate one (or very few) auditory cortico-collicular fiber. In these experiments, the LED intensity was titrated to the minimum power required for optogenetic responses to fluctuate between successful EPSPs and failures on a trial-by-trial basis (*Figure 2A and B*; mean failure rate across experiments: 44% ± 3%, n = 18 cells from N = 14 mice). The mean amplitude of successful EPSPs was generally small (0.84 ± 0.09 mV; *Figure 2C*) and similar to previous reports of unitary synapses between layer 5 pyramidal neurons in sensory cortex (*Brown and Hestrin, 2009*; *Lefort et al., 2009*). By contrast, progressively stronger LED flashes increased the amplitude of successful EPSPs compared to the minimal stimulation condition in many cells tested (n = 15 cells from N = 11 mice; *Figure 2D and E*), indicating that multiple corticofugal axons can converge onto individual shell IC neurons. The ratio of maximum to minimum EPSP showing varied >10-fold across different experiments (median: 1.67, range: 0.91–24.04, *Figure 2F*), indicating that the EPSP amplitude variability across individual neurons likely reflects the number of presynaptic auditory cortical fibers recruited during stimulation rather than differences in unitary strength. Importantly, the EPSP half-width was constant across the range of stimulus intensities (*Figure 2D*, inset; EPSP half-width ratios at maximal and threshold LED intensities: 1.04 ± 0.07). Similar results were obtained in voltage-clamp recordings, with EPSC rise times

and half-widths being similar at threshold and maximal LED intensities (*Figure 2—figure supplement 1*). Together, these results indicate that increased LED intensities recruit more axons rather than prolonging Chronos activation and temporally dispersing vesicle release from single presynaptic boutons. We thus conclude that although the strength of individual auditory cortico-collicular synapses is weak, the convergence of multiple presynaptic fibers ensures that descending signals will substantially increase shell IC neuron excitability.

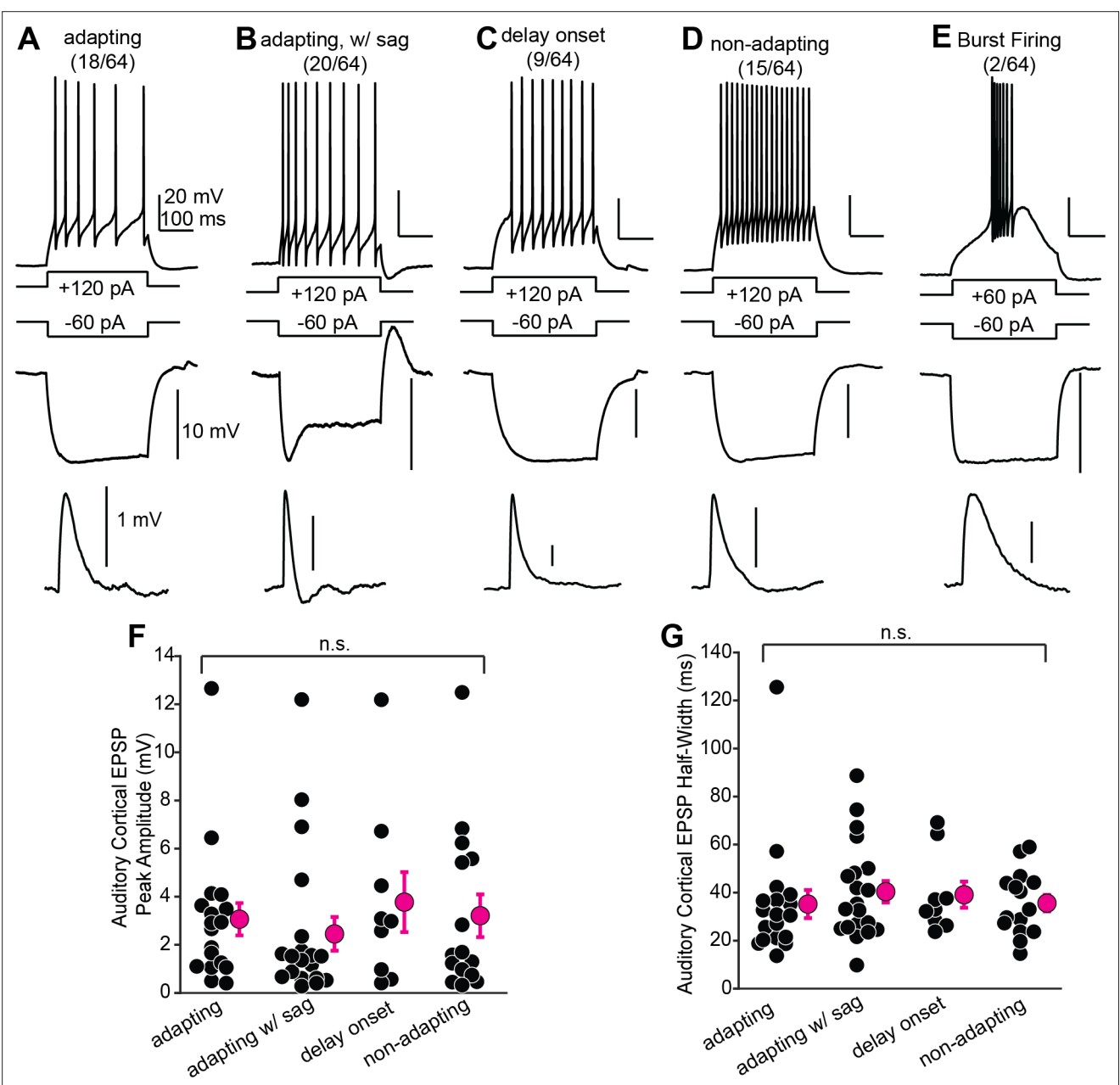

**Figure 3.** Shell inferior colliculus (IC) neurons receiving descending signals have diverse biophysical properties. (**A–E**) Examples of different shell IC neurons. Top and middle traces are spike output and membrane hyperpolarization following positive and negative current steps, respectively. Lower traces: average auditory cortical EPSPs from the same neuron. (**F, G**) EPSP peak amplitude (**F**) and half-width (**G**) as a function of biophysical category. Bursting neurons are omitted due to low n.

## Auditory cortico-collicular synapses contact shell IC neurons with diverse biophysical properties

IC neurons have a variety of firing patterns and membrane properties that potentially correspond to distinct neuronal subtypes (*Smith, 1992*; *Peruzzi et al., 2000*; *Ahuja and Wu, 2007*; *Moore and Trussell, 2017*; *Goyer et al., 2019*; *Naumov et al., 2019*; *Silveira et al., 2020*). However, whether auditory cortico-collicular synapses contact biophysically homogenous or diverse neurons is unknown. We injected 300 ms positive and negative current steps to quantify spiking patterns and passive membrane properties in a subset of our in vitro experiments described above (n = 64 cells). 62/64 cortico-recipient shell IC neurons could be classified into one of four general categories. Over half of neurons (38/64) had significant spike rate adaptation during positive current injections (*Figure 3A and B*) and qualitatively similar firing patterns as the shell IC neurons recorded in rat slices by *Smith, 1992*. However, 18 of these 'adapting' neurons responded to negative current with a sustained hyper-polarization (*Figure 3A*), whereas 20 displayed a prominent $I_h$-like 'sag' likely mediated by HCN channels (*Figure 3B*, middle trace). These data suggest a minimum of two shell IC neuron subtypes with adapting firing patterns, which can be differentiated based on the extent of $I_h$ sag. By contrast, other neurons had delayed first spikes (9/64, *Figure 3C*) or showed nonadapting discharge patterns (15/64, *Figure 3D*). Finally, 2/64 cells displayed a strikingly distinct phenotype, with a burst of spikes riding atop a 'hump'-like depolarization similar to neurons expressing T-type $Ca^{2+}$ channels (*Figure 3E*). EPSP amplitude and half-width were similar across the four major categories (peak amplitude: p=0.45; half-width: p=0.5, Kruskal–Wallis tests) despite marked differences in electrical properties. Thus, the auditory cortex broadcasts similarly strong signals to multiple, putatively distinct IC neurons, arguing that biophysical properties alone do not predict the strength of descending synapses.

## Temporal integration of auditory cortical inputs is moderately sublinear

Auditory cortico-collicular neurons in awake mice respond to acoustic stimuli with ~10–50 Hz spike trains (*Williamson and Polley, 2019*). Given the kinetics of auditory cortico-collicular EPSPs (*Figure 1*), these spike rates are expected to result in significant temporal summation of descending signals. However, the use-dependent dynamics neurotransmitter release, as well as postsynaptic ion channels, can enforce sub- or supralinear summation that effectively dictates the temporal integration of EPSPs (*Markram et al., 1998*; *Polsky et al., 2009*). How do IC neurons integrate sustained cortical activity? We addressed this question by repetitively stimulating auditory cortico-collicular axons (10 light pulses at 20 or 50 Hz, 2 ms pulse width; *Figure 4A and B*, black traces; n = 12 neurons from N = 7 mice). We quantified temporal integration by comparing the peak EPSP amplitudes observed after each light flash in the train to the amplitudes expected from the linear summation of a single auditory cortico-collicular EPSP recorded in the same neuron (*Figure 4A and B*, magenta traces). The observed peak amplitude of the 10th EPSP in the train reached 92% ± 13% and 78% ± 16% of that expected from linear summation at 20 and 50 Hz, respectively (*Figure 4C and D*). These results argue that at the population level, auditory cortical firing rates are read out as moderately sublinear shifts of the membrane potential towards threshold. These sublinear effects were likely due to frequency-dependent synaptic depression as the trough-to-peak amplitude of individual EPSPs during train stimulation showed greater reduction at faster rates (*Figure 4—figure supplement 1*; EPSP 10/EPSP 1: 83% ± 10% vs. 39% ± 9% for 20 and 50 Hz trains, respectively; p<0.001, sign-rank test).

Auditory cortico-collicular neurons display elevated firing rates at ~20 Hz for the entirety of long-duration, complex sounds (e.g., 4 s long dynamic chord stimuli; *Williamson and Polley, 2019*). We thus wondered if descending synapses maintain transmission during sustained acoustic processing, or if synaptic depression instead limits descending signals to sound onset. Stimulating auditory cortical axons with 4 s trains of light pulses at 20 Hz (*Figure 4E*) drove fast EPSPs riding atop a steady-state, tonic depolarization (*Figure 4F*; mean amplitude of DC component during the final 1 s of stimulation: 1.18 ± 0.01 mV, p=0.013, one-sample *t*-test compared to a hypothetical value of 0. n = 10 cells from n = 7 mice). Importantly, these effects were not an artifact of directly stimulating auditory cortico-collicular nerve terminals in brain slices: a similar tonic depolarization was observed in superficial IC neurons recorded in vivo using an optic fiber positioned over auditory cortex (*Figure 4G and H*; mean amplitude of DC component: 1.5 ± 0.7 mV, n = 19 cells from n = 6 mice, p=0.53 compared to in vitro data, rank-sum test).

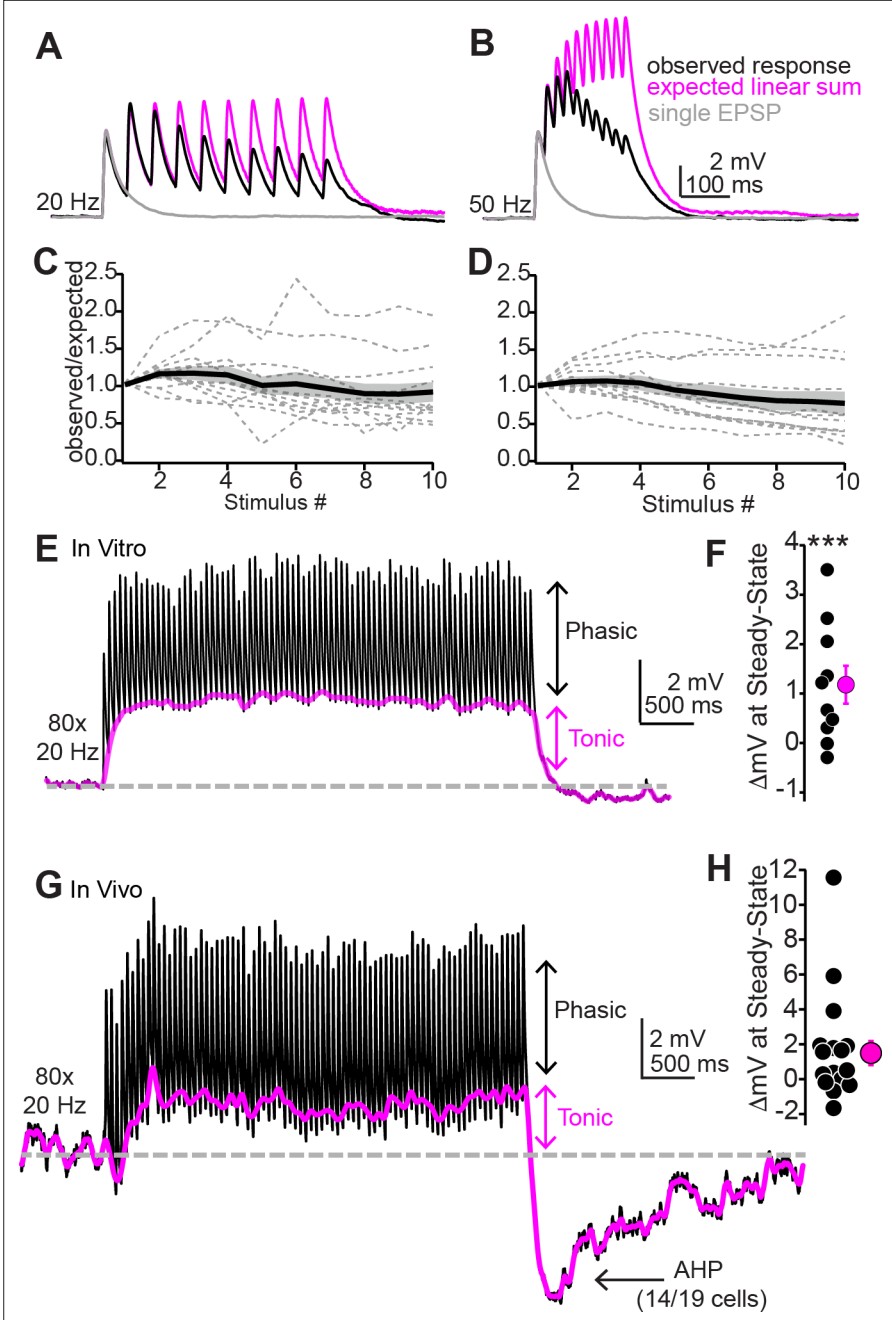

**Figure 4.** Repetitive synaptic activity tonically depolarizes shell inferior colliculus (IC) neurons despite sublinear temporal summation of EPSPs. (**A, B**) Black traces: average EPSPs evoked by 20 (**A**) or 50 Hz (**B**) trains of 10 light flashes. Gray: average EPSPs evoked by a single 2 ms light flash recorded in the same neuron. Magenta: the expected response assuming linear summation of single EPSPs at each frequency. Of note is that the recorded EPSP amplitudes in the train are smaller than the expected linear sum. Data in (**A**) and (**B**) are from the same neuron. (**C, D**) Summary of observed over expected amplitudes for EPSPs during 20 (**C**) or 50 Hz (**D**) trains. Gray dotted lines are individual neurons; black + shading are mean ± SEM. (**E**) Sustained auditory cortical transmission generates phasic EPSPs (black) riding atop a tonic depolarization (magenta) in vitro. (**F**) Group data quantifying membrane potential changes at steady state (final 1 s of stimulation). (**G, H**) Same as (**E**) and (**F**) but during in vivo recordings. Of note is the large after-hyperpolarization (AHP).

The online version of this article includes the following figure supplement(s) for figure 4:

**Figure supplement 1.** Depression of instantaneous EPSP amplitudes during trains.

**Figure supplement 2.** Large after-hyperpolarizations (AHPs) follow the cessation of auditory cortico-collicular activity in vivo.

In addition, cessation of cortical stimulation in vivo also caused a long-lasting after-hyperpolarization (AHP) in 14/19 cells: the membrane potential rapidly fell below baseline after the last stimulus ($\tau$ = 76.5 ± 1.5 ms) and recovered over several seconds ($\tau$ = 2.6 ± 0.6 s; *Figure 4—figure supplement 2*). Interestingly, this AHP was independent of postsynaptic spiking and thus may reflect buildup of feed-forward inhibition from local and long-range sources, or alternatively, a transient cessation of tonic descending excitation. Altogether, these experiments show that auditory cortico-collicular synapses can sustain transmission on seconds time scales via tonic and phasic excitation. In addition, the profound AHP following in vivo stimulation suggests that IC neuron excitability is bidirectionally yoked to auditory cortical firing patterns. Thus, increases as well as pauses in auditory cortico-collicular neuron activity may be comparably significant to IC neurons.

## NMDA receptors contribute to temporal integration of descending signals

Central excitatory transmission is predominantly mediated by AMPA and NMDA-type glutamate receptors, with NMDA receptors being particularly crucial for dendritic integration, associative plasticity, and learning. The auditory cortico-collicular pathway is involved in perceptual learning following monaural hearing loss (*Bajo et al., 2010*), and NMDA receptors in the avian IC shell homologue preferentially contribute to receptive fields generated by experience-dependent plasticity (*Feldman et al., 1996*). We thus asked to what extent auditory cortico-collicular synapses activate NMDA receptors in shell IC neurons. Interestingly, bath application of the AMPA/kainate receptor antagonist NBQX (10 µM) completely abolished EPSPs in all neurons tested (*Figure 5A and B*; peak amplitude control: 1.93 ± 0.69 mV; NBQX: 0.1 ± 0.03 mV, p=0.004, n = 9 cells from N = 7 mice, Wilcoxon sign-rank test), indicating that AMPA receptors mediate the overwhelming majority of synaptic depolarization at descending synapses.

By contrast, although the NMDA receptor antagonist R-CPP (5 µM) had minimal effect on the peak amplitude of auditory cortico-collicular EPSPs evoked with single light flashes (*Figure 5C and D*; n = 9 cells from N = 6 mice; control: 3.3 ± 0.9 mV, R-CPP: 3.4 ± 0.9 mV, p=0.67, paired *t*-test), R-CPP caused a modest but significant reduction in the EPSP half-width (*Figure 5C and E*; control: 35.7 ± 4.4 ms, R-CPP: 30.4 ± 4.7 ms; 17.6% reduction, p=0.016, paired *t*-test). This modest effect of R-CPP in current-clamp recordings was not due to a complete absence of NMDA receptors at descending synapses, nor was it due to the presence of R-CPP-resistant NMDA receptors (*Feng et al., 2005*): depolarizing neurons to +30 to +45 mV in voltage-clamp (using a $Cs^+$-rich internal solution) slowed the weighted decay time constant of optogenetically evoked EPSCs by 8.2-fold compared to negative holding potentials (*Figure 5—figure supplement 1A and B*; –60 to –70 mV: 6.1 ± 0.9 ms, + 30 to + 40 mV: 50.0 ± 0.8 ms, n = 5 cells from N = 3 mice), consistent with the biophysical properties of NMDA receptors. Furthermore, R-CPP markedly accelerated the decay tau of EPSCs recorded at positive potentials (*Figure 5—figure supplement 1C and D*; n = 7 cells from N = 4 mice; control: 70.1 ± 17.6 ms, R-CPP: 10.2 ± 1.3 ms, p=0.0156, sign-rank test). The remaining fast EPSC was abolished by NBQX (n = 3 cells from N = 2 mice), indicating that at the concentration applied (5 µM), R-CPP fully saturates NMDA receptors at auditory cortico-collicular synapses. We conclude that although AMPA receptors mediate the bulk of descending transmission at resting membrane potentials, glutamate released from descending synapses nevertheless reaches synaptic NMDA receptors to shape EPSP kinetics. Thus, although NMDA receptors contribute little to the peak amplitude of EPSPs during sparse stimulation, they may nevertheless control the integration of repetitive activity across time.

Accordingly, R-CPP significantly reduced summation of auditory cortico-collicular EPSPs evoked by a train of 10 stimuli at 50 Hz (*Figure 5F and G*; n = 10 cells from N = 8 mice; main effect of drug condition in a two-way repeated-measures ANOVA; F(1,9) = 5.91, p=0.038), resulting in a 26% reduction in the cumulative integral of the synaptic depolarization (*Figure 5F and H*; control: 1.42 ± 0.35 mV * s, R-CPP: 1.03 ± 0.26 mV * s; p=0.012, paired *t*-test). These data show that NMDA receptors prolong the integration time window for descending signals, thereby boosting postsynaptic depolarizations during sustained cortical activity.

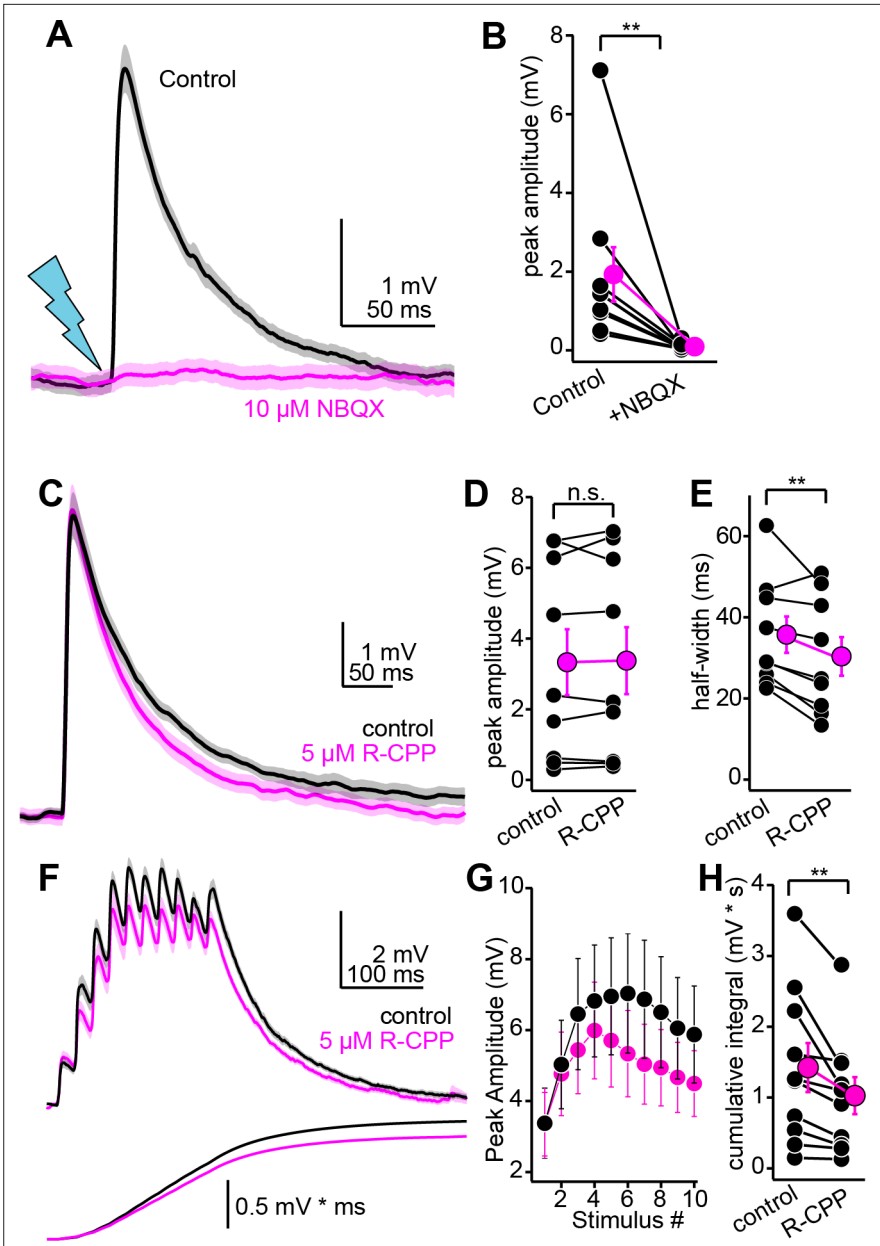

**Figure 5.** Pharmacology of descending transmission. (**A**) EPSPs (mean ± SEM) from a single neuron before (black) and after (magenta) bath application of NBQX (10 µM). Of note is the absence of residual synaptic depolarization after NBQX. (**B**) Summary data. For the effect of NBQX on descending EPSPs. (**C**) EPSPs before and after R-CPP (black and magenta traces, respectively). (**D, E**) Group data for the effect of R-CPP on EPSP peak amplitude (**D**) and half-width (**E**). Asterisks denote statistical significance. (**F**) Upper panel: EPSPs (average ± SEM) evoked by 10 light flashes at 50 Hz. Black and magenta are in control and R-CPP. Lower panel is the cumulative integral of the waveforms. (**G**) Group data showing amplitude of each EPSP in the train (mean ± SEM) before and after R-CPP (black and magenta, respectively). (**H**) Group data for the effect of R-CPP on voltage integral as in (**F**). Asterisks denote significance.

The online version of this article includes the following figure supplement(s) for figure 5:

**Figure supplement 1.** The slow component of auditory cortico-collicular excitatory postsynaptic currents (EPSCs) recorded at positive potentials is blocked by 5 µM R-CPP, consistent with the presence of R-CPP-sensitive NMDARs.

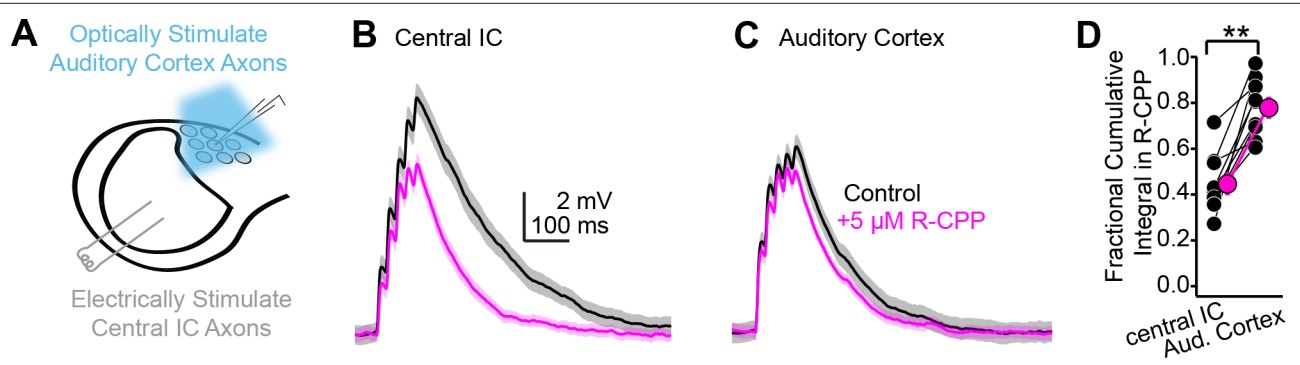

**Figure 6.** Differential contribution of NMDA receptors at ascending and descending synapses. (**A**) Cartoon of experiment: blue light flashes delivered through the microscope objective activate descending auditory cortico-collicular axons; a bipolar electrode in the central inferior colliculus (IC) stimulates ascending central -> shell IC axons. (**B, C**) EPSPs evoked by 5 × 50 Hz stimulation of central IC (**B**) or auditory cortex (**C**) axons, in the absence or presence of R-CPP (black and magenta, respectively). Data in panels (**B**) and (**C**) are from the same neuron. (**D**) Group data quantifying the fractional EPSP remaining in R-CPP (as measured by the cumulative integral of the EPSP waveform) for central IC and auditory cortico-collicular synapses.

The online version of this article includes the following figure supplement(s) for figure 6:

**Figure supplement 1.** R-CPP has a greater effect on intra-collicular EPSPs evoked with single shock compared to descending EPSPs evoked with single light flashes.

## Differential contribution of NMDA receptors at ascending and descending synapses

The modest contribution of NMDA receptors at descending synapses is surprising as previous studies suggest a rather prominent NMDA component at excitatory synapses in the IC (*Smith, 1992*; *Wu et al., 2004*). We thus wondered if our results reflected a relative paucity of NMDA receptors at all excitatory synapses onto shell IC neurons, or rather a unique feature of auditory cortico-collicular synapses. Shell IC neurons receive a prominent intra-collicular projection from the central IC, which likely transmits a significant amount of ascending acoustic information (*Saldaña et al., 1996*; *Sun and Wu, 2009*). We tested if the NMDA component differed between ascending and descending EPSPs in the same neuron using a dual-pathway stimulation approach in vitro: a bipolar stimulating electrode was positioned in the central IC and Chronos was expressed in auditory cortex to activate ascending and descending synapses, respectively (*Figure 6A*). 2.5–5 µM gabazine was present in all experiments to isolate excitatory transmission. Repetitive stimulation of either pathway (5×, 50 Hz) led to summating EPSPs that were differentially sensitive to NMDA receptor blockade (*Figure 6B and C*): 5 µM R-CPP reduced the cumulative integral of central IC and auditory cortical EPSPs to 44% ± 4% and 78% ± 4% of baseline, respectively (*Figure 6D*, n = 9 cells from n = 8 mice, p=0.0004, paired *t*-test), indicating that NMDA receptors contributed less at descending compared to ascending synapses. These results were not solely due to greater contribution of NMDARs during train stimuli at ascending compared to descending synapses. Indeed, qualitatively similar results were also observed in a between-cell comparison of R-CPP effects on ascending and descending EPSPs evoked with single stimuli: R-CPP caused a significantly greater reduction in the peak amplitude of ascending EPSPs evoked with single shocks compared to the auditory cortical EPSPs evoked with single light flashes in experiments of *Figure 5C–E* (*Figure 6—figure supplement 1A and B*; fraction peak amplitude remaining in R-CPP: 75 ± 5 vs. 103% ± 4%, n = 8 cells from N = 5 mice vs. n = 9 cells from N = 6 mice for central IC and auditory cortical inputs, respectively, p<0.001, unpaired *t*-test). Similarly, the half-width of single ascending EPSPs was modestly, albeit significantly more reduced by R-CPP compared to single descending EPSPs (*Figure 6—figure supplement 1C*; fraction half-width remaining in R-CPP: 64 ± 5 vs. 82% ± 5% for central IC and auditory cortical inputs, respectively, p=0.024, paired *t*-test). Thus, the extent of NMDA receptor contribution to descending transmission reflects a synapse-specific property of auditory cortico-collicular inputs rather than the global distribution of NMDARs in shell IC neurons.

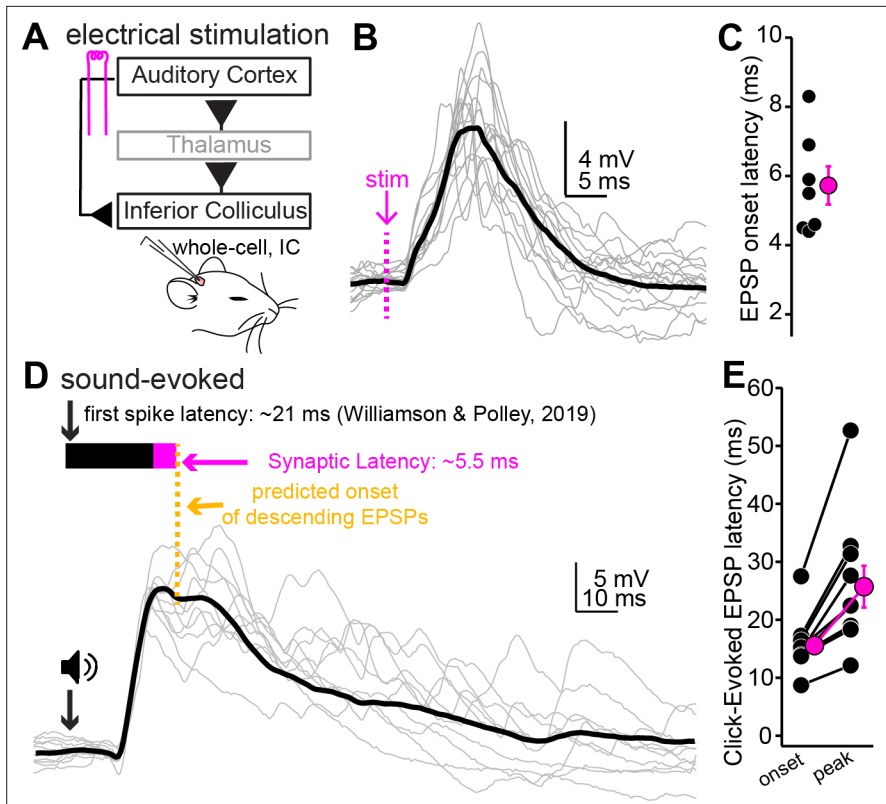

**Figure 7.** Auditory cortical feedback is predicted to collide with the peak of ascending excitation. (**A**) Cartoon of experiment. (**B**) Example recording showing short latency EPSPs following a single shock to the auditory cortex. Dotted line is onset of stimulation. Gray traces are single trials; black is average. The stimulation artifact, as well as APs riding atop the cortical EPSP, were blanked for clarity. (**C**) Group data for EPSP onset latency. (**D**) EPSPs evoked by a 200 μs click. Arrow and speaker show sound onset. Data are from a different neuron than in (**B**). Black and magenta bars above the traces show the reported first-spike latency of auditory cortico-collicular neurons from previous work and the synaptic latency calculated from panels (**B**) and (**C**), respectively. (**E**) Group data for onset and peak latency of click-evoked EPSPs.

The online version of this article includes the following figure supplement(s) for figure 7:

**Figure supplement 1.** Urethane anesthesia does not impact the onset of acoustic activity in the superficial inferior colliculus (IC).

**Figure supplement 2.** Cooling the auditory cortex impacts sound-evoked responses in the superficial inferior colliculus (IC) of awake mice.

## Auditory cortical inputs are predicted to arrive at the peak of EPSPs evoked by transient sounds

Layer 5 auditory cortico-collicular neurons in awake mice respond to sound with a mean first-spike latency of ~21 ms (*Williamson and Polley, 2019*). This value is surprisingly shorter than the reported first-spike latencies in shell IC neurons (~35 ms; *Lumani and Zhang, 2010*). Whether auditory cortical excitation arrives before, during, or after the onset of sound-evoked EPSPs in shell IC neurons effectively determines how ascending and descending signals integrate at the single-cell level, but the relative timing of distinct inputs onto IC neurons is unknown. We thus quantified the relative timing of ascending sound-evoked and descending cortical EPSPs using in vivo whole-cell recordings from superficial IC neurons of anesthetized mice. We first determined the onset latency of descending EPSPs in IC neurons using electrical stimulation of the auditory cortex (*Figure 7A*). We employed electrical rather than optogenetic stimulation for these experiments because spike onset following optogenetic stimulation is limited by the cell's membrane time constant and effective spike threshold, whereas electrical stimulation bypasses somatodendritic depolarization by directly triggering axonal spikes. Single shocks delivered to the auditory cortex evoked EPSPs with an onset latency of 5.4 ±

0.6 ms (n = 7 cells from n = 4 mice; mean depth of recorded neurons 188 ± 14 µm, *Figure 7B and C*), indicating that descending information reaches IC neurons within a few ms of AP initiation in auditory cortex. Since the mean first-spike latency of auditory cortico-collicular neurons is ~21 ms (*Williamson and Polley, 2019*) and the synaptic latency is 5–8 ms (*Figure 7B and C*), these data collectively argue that that cortical feedback begins to excite IC neurons < 30 ms after sound onset. Furthermore, assuming an axon path length of ~8 mm from auditory cortex to shell IC (*Llano et al., 2014*) and a synaptic release delay of ~2 ms, the data suggest a minimum conduction velocity of ~2.35 m/s under these conditions. These values are similar to conduction velocity estimates for the myelinated axons of layer 5 pyramidal neurons in rodents (2.9 m/s; *Kole et al., 2007*), suggesting that layer 5 neurons, and not unmyelinated layer 6 neurons, are the dominant source of descending signals to the IC.

We next calculated the timing of sound-evoked excitation onto superficial IC neurons using broadband 'click' transients (*Figure 7D*, n = 10 cells from N = 7 mice; mean depth: 204 ± 23 µm). Although sound-evoked EPSPs were typically subthreshold under our conditions, click stimuli occasionally drove spikes in 5/10 experiments (33.8% ± 16.6% of trials), with mean first-spike latencies of 29.4 ± 7.6 ms. These values are similar to previous studies in anesthetized animals showing that sound-evoked spiking of shell IC neurons is sparse (*Lumani and Zhang, 2010*; *Geis et al., 2011*; *Valdés-Baizabal et al., 2021*). Sound-evoked EPSPs had mean onset and peak latencies of 15.5 ± 1.5 and 25.7 ± 3.6 ms, respectively (*Figure 7E*), indicating that cortical feedback lags the rising phase of sound-evoked excitation by a mere 10–15 ms. These conclusions are unlikely to reflect idiosyncratic artifacts of anesthesia as the onset of click-evoked local field potentials (LFPs) was similar in awake and anesthetized mice (*Figure 7—figure supplement 1*; 9.70 ± 0.36 vs. 9.16 ± 0.32 ms in N = 11 awake and N = 8 anesthetized mice, respectively, p=0.31, unpaired *t*-test). Furthermore, although brain state can affect conduction velocity of cortical axons (*Stoelzel et al., 2017*), spike propagation is generally *faster* during high compared to low arousal levels. Our latency estimates (*Figure 7B and C*) thus likely represent an upper bound for the arrival of descending cortical signals in awake states.

The data of *Figure 7* suggest that cortical excitation arrives in IC neurons prior to, or in very close succession with, the peak of EPSPs evoked by transient sounds. Moreover, the 5–10 Hz spontaneous firing rate of layer 5 pyramidal neurons in vivo (*O'Connor et al., 2010*; *Yavorska and Wehr, 2021*) suggests that auditory cortex activity prior to sound onset could modulate the earliest phases of acoustic activity in the IC. We thus tested if auditory cortex inactivation via cooling impacts sound-evoked activity in the superficial IC of awake mice (*Figure 7—figure supplement 2A and B*).

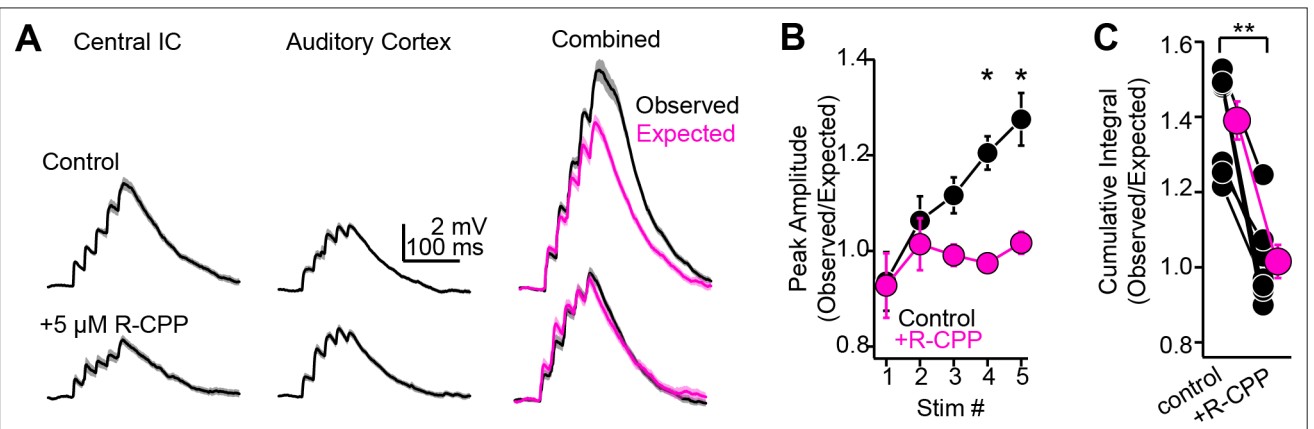

**Figure 8.** NMDA receptor-dependent, supralinear pathway integration. (**A**) Upper and lower traces are during baseline conditions and after bath applying R-CPP. Left and middle panels: EPSPs during electrical or optogenetic stimulation of central inferior colliculus (IC) or auditory cortical axons, respectively (5×, 50 Hz stimuli). Right panel ('combined'): black traces are the observed depolarization during simultaneous stimulation of both pathways. Magenta is the depolarization expected from the arithmetic sum of the waveforms following stimulation of either pathway alone (e.g., left and middle traces). Of note, the observed depolarization under control conditions is larger than expected from linear summation; blocking NMDA receptors linearizes pathway integration (compare black and magenta traces in R-CPP). (**B**) Group data plotting the ratio of observed and expected peak amplitude for each of five EPSPs in a 50 Hz train during synchronous activation of central IC and auditory cortical synapses. Asterisks denote statistical significance of Bonferroni post-hoc test for the fourth and fifth stimuli following a main effect of drug condition (p=0.038, F(1,9), two-way repeated-measures ANOVA). (**C**) Group data, observed over expected ratio of the cumulative integral during combined pathway activation in control conditions and in the presence of R-CPP. Asterisks denote statistical significance (sign-rank test).

Accordingly, bilateral cooling of the auditory cortices using Peltier devices reversibly reduced the slope of click-evoked LFPs (*Figure 7—figure supplement 2C and E*; n = 11 recordings in N = 6 mice; one-way repeated-measures ANOVA, $F_{(1.402,14.02)}$ = 13.89, p=0.0011). In addition, 9 out of these 11 recordings also had measurable multiunit spike activity whose rates increased following click sounds (*Figure 7—figure supplement 2B and D*). Consistent with the LFP data, the onset of click-evoked multiunit activity was reversibly delayed by auditory cortical cooling (*Figure 7—figure supplement 2D and F*; n = 9 recordings in N = 6 mice; one-way repeated-measures ANOVA, $F_{(1.294,10.35)}$ = 26.15, p=0.0002). Altogether, these results indicate that auditory cortical activity substantially impacts how IC neurons respond to transient sounds.

## NMDA receptor-dependent, supralinear pathway integration in shell IC neurons

Our latency measurements (*Figure 7*) suggest that ascending information is rapidly followed by descending cortical excitation. This temporal overlap is intriguing because ascending and descending synapses express NMDA receptors (*Figure 6*), which in other cell types enable cooperative interactions between coactive pathways onto the same neuron (*Takahashi and Magee, 2009*). We thus hypothesized that appropriately timed cortical feedback might integrate nonlinearly with ascending inputs from the central IC, thereby generating a synaptic depolarization larger than expected from the sum of either pathway active in isolation. We tested this idea in vitro using our dual-pathway stimulation approach (*Figure 6*) while recording from shell IC neurons. We first recorded the synaptic depolarization following stimulation of ascending and descending pathways in isolation (five stimuli at 50 Hz; *Figure 8A*, upper traces). We next simultaneously activated the two pathways such that the onset of cortical EPSPs collided with the peak of ascending EPSPs, as predicted from our in vivo latency measurements (*Figure 7*). The observed depolarization during synchronous pathway activation was on average significantly larger than expected from the arithmetic sum of each pathway stimulated alone (*Figure 8A*; n = 12 cells from N = 9 mice; cumulative integral observed: 1.90 ± 0.35 mV * ms, expected: 1.45 ± 0.22 mV * ms, p=0.007, paired *t*-test test), indicating that coincident activity of ascending and descending pathways summates supralinearly. Importantly, R-CPP mostly abolished this supralinear summation, such that the depolarization during coincident activation in R-CPP now equaled the expected sum of each pathway activated alone (*Figure 8B and C*, lower traces; n = 7 cells from N = 7 mice; observed/expected control: 1.39 ± 0.05; in R-CPP: 1.02 ± 0.04, p=0.0156, sign-rank test). Thus, synaptic NMDA receptors provide a supralinear boost when descending excitation follows ascending activity, thereby promoting the nonlinear mixing of distinct pathways in single IC neurons.

## Discussion

We have shown that the majority of neurons in the superficial (shell) IC layers receive reasonably strong excitation from auditory cortex that is predicted to arrive within 30 ms following sound onset. Given the relatively long latency of sound-evoked PSPs in shell IC neurons (*Figure 7D and E*; see also *Valdés-Baizabal et al., 2021*), we suggest that descending signals arrive near the peak of ascending excitation evoked by short sounds. However, our measurements of corticofugal latencies likely represent an upper bound for descending transmission as our experiments were performed under anesthesia; the conduction velocity of cortical axons is generally increased in alert compared to nonalert animals (*Stoelzel et al., 2017*) such that corticofugal excitation may arrive at the IC substantially faster in awake, behaving animals.

Cooling the auditory cortex slowed the rising slope of sound-evoked LFPs in awake mice, as originally observed by *Nwabueze-Ogbo et al., 2002* using pharmacological inactivation in anesthetized rats. In addition, cortical cooling substantially delayed the onset of sound-evoked spiking in the IC. A potential explanation is that in addition to rapid feedback following sound onset, the behaviorally modulated, ~5–10 Hz spontaneous firing of layer 5 corticofugal neurons (*O'Connor et al., 2010*; *Yavorska and Wehr, 2021*) may predictively dictate how IC neurons respond to incoming brainstem signals. Indeed, descending transmission is maintained during extended activity periods (*Figure 4*), which may allow the cortex to tonically depolarize IC neurons as a function cortical state. However, an important caveat is that our cooling approach also likely impacted corticofugal transmission to the cochlear nuclei and superior olive. Although subtectal corticofugal projections are comparatively

sparser than the auditory cortico-collicular pathway (*Weedman and Ryugo, 1996*; *Doucet et al., 2003*; *Coomes and Schofield, 2004*), a formal possibility nevertheless remains that some of our observed effects are due to modulation of ascending brainstem inputs to the IC.

Our data suggest that shell IC neurons are the major target of auditory cortico-collicular fibers. However, the nonuniform thickness of the IC shell across medial-lateral axis (*Barnstedt et al., 2015*) indicates that we cannot rule out that some of our in vivo data are from neurons in the most dorsal region of the central IC. Indeed, in vivo intracellular recordings from neurons in the deep IC layers report EPSPs following auditory cortex stimulation (*Mitani et al., 1983*; *Qi et al., 2020*), implying that some functional cortical synapses may in fact target central IC neurons. However, in vitro circuit mapping experiments imply that monosynaptic connections between auditory cortex and central IC may be rare (*Xiong et al., 2015*; *Song et al., 2018*), and our trans-synaptic labeling and in vivo electrophysiology data support the conclusion that shell IC neurons process the bulk of descending signals. However, more thorough functional circuit-mapping studies are needed to directly quantify the extent and synaptic potency of descending inputs in specific IC subdivisions.

Interestingly, a recent study suggested that vasoactive intestinal peptide (VIP)-expressing GABAergic interneurons in auditory cortex send axonal projections to the shell IC (*Bertero et al., 2021*). In our conditions, the AMPA/kainate receptor antagonist NBQX largely abolished descending transmission (*Figure 5*); we did not observe any inhibitory postsynaptic potentials (IPSPs) in the presence of NBQX, as would be expected from the direct stimulation of GABAergic axons. However, these newly identified corticofugal GABAergic synapses have low release probability, signal mainly via 'spillover' transmission (*Isaacson et al., 1993*; *Szabadics et al., 2007*), or operate via the release of VIP that would be difficult to quantify with our methods. Future studies are required to disentangle the relative contributions of glutamatergic and GABAergic corticofugal transmission in the IC.

## Synapse-specific contribution of NMDA receptors in shell IC neurons

The minor contribution of NMDA receptors to auditory cortico-collicular transmission is somewhat surprising as synaptic NMDA receptors are activated even at hyperpolarized membrane potentials in central IC neurons (*Ma et al., 2002*; *Wu et al., 2004*; *Goyer et al., 2019*; *Kitagawa and Sakaba, 2019*). However, our results do not simply reflect a global paucity of NMDA receptors at excitatory synapses onto dorso-medial shell IC neurons, but rather can be explained by a pathway-specific contribution of specific glutamate receptor subtypes to the synaptic depolarization: NMDA receptor blockade reduced EPSPs to a greater extent at central IC -> shell IC compared to auditory cortico-collicular synapses. Although a simple explanation is that the total number of synaptic NMDA receptors in shell IC neurons differs in a pathway-specific manner, we cannot exclude differences in NMDA receptor subunit composition (*Schwartz et al., 2012*), glutamate diffusion (*Arnth-Jensen et al., 2002*), and synapse location (*Branco et al., 2010*; *Song et al., 2018*) as contributing factors.

## Nonlinear integration of ascending and descending signals

Intra-collicular synapses originating from the central IC likely provide a significant amount of ascending acoustic input to shell IC neurons. Indeed, reported first-spike latencies of central IC neurons typically lead the onset of sound-evoked EPSPs in shell IC neurons (*Syka et al., 2000*; *Hurley and Pollak, 2005*), and central IC neurons send tonotopically organized axonal projections to the IC shell that conspicuously mirror the tonotopic distribution of best frequencies in the shell IC (*Saldaña and Merchán, 1992*; *Wong and Borst, 2019*). By contrast, the onset of sound-evoked EPSPs in shell IC neurons typically begins prior to the reported first spikes of auditory cortico-collicular neurons, thereby ruling out the possibility that acoustic responses are solely inherited from auditory cortex. Instead, axonal conduction velocities along ascending and descending pathways impose an obligatory delay such that cortical feedback excitation arrives ~25–30 ms following the onset of sound-evoked EPSPs. Notably, while glutamate released from ascending terminals will have unbound and diffused away from low-affinity synaptic AMPA receptors prior to the onset of cortical feedback excitation (*Clements et al., 1992*), high-affinity NMDA receptors are expected to remain bound with glutamate during this time (*Lester and Jahr, 1992*). Extracellular $Mg^{2+}$ imparts a voltage dependence to the NMDA receptor channel; the additional depolarization provided by cortical feedback would thus be expected to cooperatively enhance current flowing through NMDA receptors at ascending synapses active immediately prior to the onset of cortical feedback. This prediction is supported by

our data showing a NMDA receptor-dependent, supralinear summation of ascending and descending inputs that are activated similar to their expected timing in vivo. These single-cell, biophysical operations can potentially explain the nonlinear changes in IC neuron receptive fields during cortical inactivation (*Yan and Suga, 1999*; *Nakamoto et al., 2008*; *Nakamoto et al., 2010*). An important avenue for future research will be to determine how other voltage-gated channels contribute to integrative nonlinearities in IC neurons. Indeed, although the NMDA receptor antagonist R-CPP largely abolished our observed effects, we did not formally investigate the role of other dendritic conductances such as Na$^+$ (*Stuart and Sakmann, 1995*; *Apostolides and Trussell, 2014*; *Hsu et al., 2018*) or Ca$^{2+}$ channels (*Takahashi and Magee, 2009*; *Fletcher and Williams, 2019*) in synaptic integration.

NMDA receptor-dependent nonlinearities are typically thought of as unique features of cortical pyramidal neurons that support the computational power of these high-level microcircuits. Nevertheless, multiplicative integration of sound localization cues has been observed in single neurons of barn owl IC (*Pena and Konishi, 2001*; *Pena and Konishi, 2002*), and the NMDA receptor-dependent nonlinearity we observe is comparable in magnitude to that reported during clustered activation of neighboring synapses in CA1 neurons (*Harnett et al., 2012*). Together, these data suggest that cooperative interactions between temporally correlated inputs may be a common neuronal operation throughout the central nervous system. An important distinction, however, is that most, if not all, excitatory synapses in pyramidal neurons reside on dendritic spines; this compartmentalization greatly limits any cooperative interactions to neighboring synaptic inputs on the same branch (*Gasparini and Magee, 2006*; *Losonczy and Magee, 2006*). By contrast, auditory cortico-collicular axons often form large (~5 μm$^3$) synapses on the soma of dorsal IC neurons (*Song et al., 2018*). Depending on the impedance mismatch between the somatic and dendritic compartments, synaptic depolarizations at the soma could propagate passively throughout the neuron's multiple dendrites, thereby enabling cortical signals to nonlinearly control ascending information irrespective of the spatial relationship of coactive inputs. However, further studies are necessary to identify the precise anatomical relationship between ascending and descending synapses in single IC neurons. Finally, we did not observe any overt correlation between the strength of descending EPSPs and the diverse biophysical properties of shell IC neurons. However, an important consideration is that our metrics of neuronal diversity perhaps do not reflect explicit cell-type categories in the IC. Future studies comparing descending transmission as a function of neurotransmitter phenotype (*Naumov et al., 2019*) or molecular markers (*Goyer et al., 2019*; *Silveira et al., 2020*; *Kreeger et al., 2021*) will likely provide more nuanced insight into the diversity of corticofugal control of the IC.

## Implications for predictive control of tectal activity, synaptic plasticity, and perceptual learning

The properties of corticofugal synapses could enable a context-dependent modulation of IC neurons across multiple timescales. Indeed, the rapid onset of descending EPSPs following cortical spikes (~5–8 ms) is more than twice as fast as sensory-evoked cortical gamma rhythms (30–50 Hz). Thus, descending signals could effectively synchronize neural ensembles across the ascending auditory hierarchy either to the temporal envelope of sound (*Weible et al., 2020*) or to internally generated rhythms. Alternatively, rapid auditory cortico-collicular transmission may be particularly advantageous in driving innate behaviors in response to sound. Indeed, auditory cortico-collicular neuron activity, either via optogenetic stimulation or loud sounds, directly triggers escape and flight behaviors in mice; these effects likely occur via a descending activation of shell IC neurons projecting to the PAG (*Xiong et al., 2015*). As such, cortical signals could potentially trigger evolutionarily conserved motor programs to benefit survival.

We also found that descending synapses sustained transmission and drove tonic depolarizations even during seconds-long activity patterns, such that IC neurons may also integrate slower cortical state fluctuations. Intriguingly, auditory cortical neurons in behaving animals show enhanced firing rates during the delay period of auditory working memory tasks (*Gottlieb et al., 1989*), which apparently precedes similar activity patterns in prefrontal cortex (*Huang et al., 2016*). If these working memory-related neuronal ensembles include auditory cortico-collicular neurons, sustained transmission from descending synapses could cause seconds-long increases in IC neuron excitability based on working memory content. Accordingly, persistent delay period activity is observed in ~10% of IC neurons when rats engaged in an auditory working memory task (*Sakurai, 1990*), although future

studies are necessary to determine the extent to which this activity is inherited from descending auditory cortical pathways.

Several studies now show that layer 5 corticofugal pyramidal neurons are necessary for perceptual learning in multiple different sensory tasks. Optogenetic inhibition of layer 5 pyramidal neurons in somatosensory cortex prevents behavioral adaptation following cue-related changes in a tactile detection task, although the same manipulation had no effect on touch perception (*Ranganathan et al., 2018*). Similarly, lesioning visual corticostriatal neurons prevents acquisition, but not performance of a visual detection task (*Ruediger and Scanziani, 2020*). In the auditory system, chemical lesions of auditory cortico-collicular neurons prevent the experience-dependent recovery of sound localization following monaural hearing loss (*Bajo et al., 2010*), although auditory cortex becomes dispensable once animals have learned to localize sounds using monaural cues (*Bajo et al., 2019*). Thus, although necessary for perceptual learning, corticofugal synapses may not be the primary locus of experience-dependent plasticity. Indeed, classic studies in barn owls suggest that ascending central IC -> external (shell) IC synapses are the first site of experience-dependent, spatial map plasticity in the auditory system (*Brainard and Knudsen, 1993*). In tandem with our current study, these results suggest that auditory cortico-collicular synapses' contributions to perceptual learning may not lie in their explicit ability to undergo classical Hebbian associative plasticity, but rather as permissive forces of heterosynaptic plasticity at ascending synapses.

## Materials and methods
### Surgery for viral injections
All experiments were approved by the University of Michigan's IACUC and performed in accordance with NIH's guide for the care and use of laboratory animals. All surgical procedures were performed under aseptic conditions. Surgeries were performed on 4–7-week-old male or female C57BL6/J mice purchased from Jackson Labs or offspring of CBA × C57BL6/J matings bred in-house for electrophysiology experiments. For the trans-synaptic labeling experiments of *Figure 1—figure supplement 2*, we used 6–8-week-old Ai14 fl/fl mice bred in-house (Jackson Labs stock #007914). Mice were deeply anesthetized with 4–5% isoflurane vaporized in $O_2$ and mounted in a rotating stereotaxic frame (model 1430, David Kopf Instruments). Isoflurane was subsequently lowered to 1–2% to maintain a deep anesthetic plane, as assessed by the absence of paw withdrawal reflex and stable respiration (1–1.5 breaths/s). Body temperature was maintained near 37–38°C using a feedback controlled, homeothermic heating blanket (Harvard Apparatus). Mice were administered 5 mg/kg carprofen after induction as a pre-surgical analgesic. The scalp was clear of hair, swabbed with betadine, and a small incision was made in the skin overlying the left hemisphere. Topical 2% lidocaine was then applied to the wound margins. The stereotaxic frame was rotated ~50°, allowing a vertical approach perpendicular to the layers of auditory cortex. A 200–400 µm craniotomy was carefully opened over the left auditory cortex (–2.75 mm from bregma, centered on the lateral ridge) using a 0.5 mm diameter dental burr (19007–05, Fine Science Tools) and Foredom microdrill. The skull was frequently irrigated with chilled phosphate buffered saline (PBS) to prevent overheating during drilling. Following the craniotomy, a glass pipette (0.1–0.2 mm diameter at the tip) containing the pAAV-Syn-Chronos-GFP (Addgene #59170-AAV1) or AAV1-hSyn-Cre (Addgene #105553-AAV1) virus penetrated the auditory cortex at a rate of <10 µm/s using a motorized micromanipulator. A total of 100–200 nL virus was injected at 2–4 sites 810 and 710 µm below the pial surface (25–50 nL per site). Following injections, the pipette was maintained in place for an additional 5 min before slowly retracting at a rate of <10 µm/s. At the end of the surgery, the craniotomy was filled with bone wax, the skin was sutured, and the mouse was removed from the stereotax. Immediately following surgery, mice were given an analgesic injection of buprenorphine (0.03 mg/kg, s.c.) and allowed to recover on a heating pad before returning to their home cage. An additional postoperative dose of carprofen was administered 24 hr following surgery.

### In vivo electrophysiology
2–4 weeks following viral injections, mice were deeply anesthetized with isoflurane and mounted in a stereotax as described above. The skin overlying the skull was removed, the left temporal muscle was retracted, the stereotax was rotated ~50°, and a 2–2.5 mm craniotomy was carefully opened over the left auditory cortex. For optogenetic stimulation in *Figure 1*, the dura overlying the auditory cortex

was left intact and a cranial window was implanted over the exposed brain with cyanoacrylate glue and dental cement. For the electrical stimulation experiments in *Figure 7*, a small slit was carefully made in the dura and the craniotomy was subsequently sealed with silicone elastomer. The stereotaxic frame was returned to the horizontal position and a custom titanium headbar was affixed to the skull with dental cement. A 300–500 μm craniotomy was opened over the left IC and filled with a silicone elastomer plug. The mouse was then removed from the stereotax, anesthetized with urethane (1.5 g/kg, i.p.), and head-fixed in a custom-made sound attenuation chamber. Body temperature during the experiment was maintained at 37–38°C with a custom-designed, feedback-controlled heating blanket. For optogenetic stimulation, a 0.5 NA, 400 μm core optic fiber (Thorlabs M45L02) coupled to a 470 nm LED (Thorlabs M470F3) was mounted on a micromanipulator and positioned <1 mm away from the auditory cortex cranial window. For electrical stimulation experiments, the silicone plug over auditory cortex was removed and a bipolar platinum-iridium electrode (FHC 30210) was carefully inserted ~800 μm into auditory cortex at an angle roughly perpendicular to the cortical layers. Electrical stimuli were delivered via a custom stimulus isolator designed in house. Sound clicks (0.2 ms duration) were presented at ~91 dB peak equivalent SPL via a free-field speaker (Peerless XT25SC90-04) positioned ~10 cm from the mouse's right ear. For whole-cell recordings, the silicone plug over the IC was removed and patch-clamp recordings were obtained from IC neurons via the 'blind patch' approach using pipettes filled with $K^+$-rich internal solution containing (in mM): 115 K-gluconate, 4 KCl, 0.1 EGTA, 10 HEPES, 14 Tris-phosphocreatine, 4 Mg-ATP, 0.5 Tris-GTP, 4 NaCl, pH 7.2–7.3, 290 mOsm (open tip resistance: 5–10 MΩ). Data were acquired using an AM Systems model 2400 patch-clamp amplifier, online filtered at 2–10 kHz, and digitized at 50 kHz with a National Instruments PCI-6343 card + BNC2090A interface controlled by MATLAB-based acquisition software (Wavesurfer). Data were recorded with the amplifier's pipette capacitance neutralization circuitry activated. Series resistance was typically between 20 and 60 MΩ. Field potentials in *Figure 7—figure supplement 1* were recorded with saline-filled glass pipettes lowered 200–250 μm into the IC.

## Auditory cortical inactivation by cooling

We built battery-operated, constant current-driven thermoelectric cooling devices designed around a small Peltier module (NL1020T-01AC, Marlow Industries). The hot side of the Peltier chip was mounted to a 6.35 mm diameter copper rod (length: ~ 76 mm) using thermally conductive adhesive. Fin-type heatsinks were mounted to the copper rod. A machined blunt copper pin (3 mm at the base, 2 mm at the tip) was mounted on the cold side using the same adhesive and made contact with the dura mater over the auditory cortices. We performed control experiments in anesthetized mice (n = 5 cooling attempts in N = 4 mice) to verify that our devices effectively cooled the auditory cortices while minimally affecting the IC (Δtemperature in IC during cortical cooling = −1.9 ± 0.6°C). The settings used during our recordings bilaterally reduced the temperature of deep cortical layers to 14–17°C, which suffices to largely abolish auditory cortical activity (*Lomber et al., 1999*; *Coomber et al., 2011*; *Anderson and Malmierca, 2013*).

We used 8–12-week-old C57Bl6/J mice for these experiments. Awake mice were handled for 3–5 days prior to recording and acclimated to head fixation while sitting comfortably in a PLEXIGLAS tube. Following acclimation, mice were anesthetized, an ~0.5 mm craniotomy was opened over the left IC, and 2 mm craniotomies were opened over the left and right auditory cortices. The craniotomies were sealed with silicone elastomer and the mouse was allowed to recover for ~2 hr prior to recording. For recording, the silicone plugs were removed, the copper pin of the Peltier devices was positioned in contact with the left and right auditory cortices, and the craniotomies were covered with 3–4% agar in saline. A saline-filled glass electrode (~1–2 MOhm open tip resistance) was lowered into the superficial IC (~200 μm from surface) to record click-evoked field potentials and multiunit clusters before, during, and after cooling of auditory cortices. After the recording, the craniotomies were sealed with silicone and the mouse was returned to its homecage. Each subject underwent 1–3 recording sessions.

## In vitro electrophysiology

2–4 weeks following viral injections, mice were deeply anesthetized with isoflurane, swiftly decapitated, and the brains carefully removed in warm (~34°C), oxygenated ACSF containing (in mM) 119 NaCl, 25 $NaHCO_3$, 3 KCl, 1.25 $NaH_2PO_4$, 15 glucose, 1 $MgCl_2$, 1.3 $CaCl_2$, 1 ascorbate, 3 pyruvate.

200–300-µm-thick coronal slices of the IC were prepared with a vibratome (Campden Instruments). On each slice, a small cut was made in the lateral portion of the right cerebellum or right IC to aid with visual identification of the uninjected hemisphere. Slices were then incubated at 34°C in a holding chamber-filled ACSF for 25–30 min and subsequently stored at room temperature. Experiments were generally performed within 3–4 hr following slice preparation. Following incubation, a slice was transferred to a recording chamber and held in place with single strands of unwaxed dental floss tightly strung around a platinum 'harp.' The slice was continuously perfused with oxygenated ACSF heated to 32–34°C (2–4 mL/min; chamber volume: ~ 1 mL). 2–5 µM SR95531 was added to the ACSF to block GABA$_A$ receptors in most experiments of *Figures 5C–H and 6*, *Figure 6—figure supplement 1*, *Figure 8*, all voltage-clamp experiments, and some experiments of *Figure 1*. Neurons in the dorsomedial shell IC were visualized via DIC or Dodt contrast optics using a ×40 or ×63 objective (Zeiss Axioskop 2 FS Plus or Olympus BXW51 microscope). Neurons were targeted for whole-cell current-clamp recordings with pipettes filled with the same K$^+$-rich internal solution used for in vivo recordings (open tip resistance: 3–6 MΩ). For whole-cell voltage-clamp experiments in *Figure 2—figure supplement 1*, EPSCs were recorded at holding potentials between –60 and –70 mV with our standard K$^+$ internal solution or a Cs$^+$ based solution containing (in mM): 110 cesium methanesulfonate, 10 QX-314-Bromide, 0.1 EGTA, 10 HEPES, 0.5 Tris-GTP, 4.5 MgATP, 5 TEA-Cl, 10 Tris-phosphocreatine. Voltage-clamp experiments requiring positive holding potentials were exclusively performed using the Cs$^+$-based recipe. In some experiments, 30 µM Alexa 594 or 0.1% biocytin were added to the internal solution to visualize neuronal morphology via online fluorescence or post-hoc histological reconstruction.

Data were acquired with a Multiclamp 700B or AM Systems model 2400 amplifier, online filtered at 2–10 kHz, and digitized at 50 kHz with a National Instruments PCI-6343 card + BNC2090A interface controlled by Wavesurfer. In current-clamp, pipette capacitance neutralization was employed and bridge balance was maintained (series resistance typically 10–30 MΩ). In a few instances, a small amount of negative bias current (–5 to –50 pA) was injected to hyperpolarize neurons and prevent spike initiation during optogenetic activation of cortico-collicular synapses. Series resistance compensation was employed in voltage-clamp experiments (60–80%, bandwidth: 3 kHz). For dual-pathway experiments, the central IC was electrically stimulated using an AM Systems model 2100 stimulus isolator delivering mono- or biphasic shocks to a theta glass bipolar electrode placed ~500 µm from the recorded neuron. In these experiments, we made an effort to titrate the stimulation strength such that separate and combined pathway stimulation did not elicit spikes on most trials. Drugs were obtained from Tocris or HelloBio, aliquoted as stock solutions in distilled water, and stored at –20°C until the day of the experiment.

## Data analysis

Electrophysiology data were analyzed using custom MATLAB scripts. EPSP analyses were performed on averages of multiple trials (typically >10 trials per condition) after baseline membrane potential subtraction and lowpass filtering at 1 kHz unless explicitly noted in the text. Peak amplitudes of single EPSPs were calculated by averaging data points ± 0.1 ms around the local maximum following optogenetic stimulation Half-widths were calculated as the full-width at half-maximum of the peak. EPSP onset latency was defined as the time following optogenetic stimulus onset when the membrane potential reaches 20% of peak. The tonic EPSP amplitude during 20 Hz trains was calculated as follows: we first linearly interpolated the membrane potential data between each light flash to remove the phasic EPSP component. The trace was then smoothed using a 50 ms sliding window. Datapoints during the final 1 s period of the stimulus train were then averaged to estimate the amplitude of tonic membrane potential change. In certain experiments of *Figures 4, 7 and 8*, train stimuli and click sounds occasionally triggered APs in IC neurons both in vivo and in vitro. In these cases, APs were digitally removed prior to averaging the traces by linearly interpolating 0.1–0.2 ms of datapoints after the membrane potential crossed spike threshold (~20 mV/ms). Shock artifacts during electrical stimulation experiments were similarly removed via linear interpolation. In summary plots, black symbols are individual cells, magenta is mean ± SEM, and lines connect data from the same recording unless otherwise stated.

The expected linear waveforms for temporal summation experiments were calculated as follows. The average waveform of a single optogenetically evoked EPSP was peak normalized to the first EPSP

in the recorded 20 or 50 Hz train from the same cell. We subsequently convolved the single EPSP waveform with a 20 or 50 Hz binary pulse train using the MATLAB function convr(). We then calculated the peak amplitude ratios for each EPSP in observed and expected trains.

For the free-field sound presentation experiments of *Figure 7D and E*, we limited our analyses to superficial IC neurons that showed onset EPSPs in response to clicks. Other IC neurons encountered during these experiments showed either sound-evoked IPSPs (n = 7 cells from N = 5 mice) or IPSPs followed by rebound depolarizations (n = 6 cells from N = 5 mice); analyses of these data will be presented in a separate report. Click-evoked field potentials in *Figure 7—figure supplements 1 and 2* were analyzed after baseline subtraction and low-pass filtering the records at 500 Hz. Multiunit activity was detected as threshold crossings after applying a bandpass filter (between 300 Hz and 5 kHz) to the recordings. Peristimulus time histograms (PSTHs) of spike activity were generated with 100 µs bin-widths and smoothed with a 2 ms sliding window. Onset latencies of field potentials and multiunit PSTHs were defined as the time following click onset at which the data reached 20% of its peak. Average LFP waveforms and PSTHs were composed of 300–302 trials per condition.

In dual-pathway experiments of *Figure 8*, the onset latency of ascending and descending EPSPs varied across cells. Thus, the relative timing of electrical and optogenetic stimulation during combined pathway activation was calculated online and on a cell-by-cell basis, such that the onset of descending auditory cortical EPSPs collided with the peak of ascending EPSPs from central IC as predicted from our in vivo latency measurements (*Figure 7*; range of $\Delta t$ between stimulation of descending and ascending synapses: –1.3–16.4 ms). These stimulation parameters were held constant across control and R-CPP conditions for each cell. The expected linear summation was calculated by digitally summing the average synaptic waveforms following stimulation of either pathway alone, accounting for the temporal offset employed during synchronous pathway activation.

## Histology and confocal imaging

Mice were deeply anesthetized in a glass induction chamber circulated with 4.2 mL isoflurane and transcardially perfused with ~80–100 mL of PBS followed by ~80–100 mL of 10% buffered formalin (Fisher Scientific Cat# 23-245684). Brains were carefully removed, stored in 10% formalin, and protected from light for 24 hr. Subsequently, brains were stored in PBS for up to 72 hr and 100 µm thick coronal slices were cut using a ceramic blade (Cadence Endurium) and a Leica VT1000s vibratome, mounted onto slides and coverslipped using Fluoromount, then protected from light and allowed to dry at room temperature for ~12–24 hr. Slides were then stored at 4°C until ready for use. Images were collected using a Leica TCS SP8 laser scanning confocal microscope equipped with a ×10 objective.

## Statistics

Although not explicitly predetermined prior to data collection, sample sizes reflect commonly accepted standards in the field. Data were tested for normality using a Lilliefors test prior to statistical comparisons. Parametric, two-tailed *t*-tests were employed for normally distributed data. Nonparametric rank-sum or sign-rank tests were used when one or more of the distributions deviate from normal. Alpha was corrected for multiple comparisons in post-hoc significance tests following ANOVA. Statistics were run in MATLAB or GraphPad Prism 9.

## Acknowledgements

Funding was generously provided by the Whitehall Foundation, Hearing Health Foundation, and NIH/NIDCD R01DC019090 to PFA, as well as T32 DC005356 and T32 DC000011 pre-doctoral awards to HMO. We thank Drs. Michael Roberts, Marina Silveira, Gunnar Quass, and Meike Rogalla for critical comments on the manuscript.

# Additional information

## Funding

| Funder | Grant reference number | Author |
| --- | --- | --- |
| Whitehall Foundation | | Pierre F Apostolides |
| Hearing Health Foundation | | Pierre F Apostolides |
| National Institute on Deafness and Other Communication Disorders | R01DC019090 | Pierre F Apostolides |
| National Institutes of Health | T32 DC005356 | Hannah M Oberle |
| National Institutes of Health | T32 DC000011 | Hannah M Oberle |

The funders had no role in study design, data collection and interpretation, or the decision to submit the work for publication.

## Author contributions

Hannah M Oberle, Formal analysis, Investigation, Methodology, Writing – review and editing; Alexander N Ford, Investigation, Methodology, Writing – review and editing; Deepak Dileepkumar, Methodology; Jordyn Czarny, Investigation; Pierre F Apostolides, Conceptualization, Formal analysis, Funding acquisition, Investigation, Methodology, Project administration, Supervision, Writing – original draft, Writing – review and editing

## Author ORCIDs

Hannah M Oberle [iD] http://orcid.org/0000-0003-2062-143X
Pierre F Apostolides [iD] http://orcid.org/0000-0003-2512-8476

## Ethics

All experiments were approved by the University of Michigan's IACUC and performed in accordance with NIH's Guide for the care and use of laboratory animals.

## Decision letter and Author response

Decision letter https://doi.org/10.7554/eLife.72730.sa1
Author response https://doi.org/10.7554/eLife.72730.sa2

# Additional files

## Supplementary files

• Transparent reporting form

## Data availability

All data generated or analyzed during this study are included in the manuscript. Source Data files will be provided prior to publication.

The following dataset was generated:

| Author(s) | Year | Dataset title | Dataset URL | Database and Identifier |
| --- | --- | --- | --- | --- |
| Ford H, Dileepkumar A, Czarny D, Apostolides J | 2022 | Synaptic Mechanisms of Top-Down Control in the Non-Lemniscal Inferior Colliculus | https://doi.org/10.5061/dryad | Dryad Digital Repository, 10.5061/dryad.6djh9w12v |

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
