## [Editor Report]

This study provides evidence for NMDA-dependent nonlinear integration of corticofugal inputs with feedforward sound-driven inputs in inferior colliculus. This unexpected property will be important for understanding the role of cortical feedback in sound processing.

---

## [Decision Letter]

**Decision letter after peer review:**

Thank you for submitting your article "Synaptic Mechanisms of Top-Down Control by The Auditory Cortico-Collicular Pathway" for consideration by *eLife*. Your article has been reviewed by 3 peer reviewers, including Brice Bathellier as Reviewing Editor and Reviewer #1, and the evaluation has been overseen by Andrew King as the Senior Editor. The following individuals involved in review of your submission have agreed to reveal their identity: Ursula Koch (Reviewer #2); Manuel S. Malmierca (Reviewer #3).

Essential revisions:

1) The description of short-term plasticity in figure 4 is too superficial, and not precise enough to be used in models. Synapses seem to be depressing but not too much. Ideally one would fit the Markram Tsodyks model for synaptic depression, but that may be out of scope. The authors should at least extract better the adaptation kinetics of PSC trains, for example by reporting the ratio between first and last PSC (not the PSP) for the two repetition frequencies tested, or by measuring a decay time constant for each stimulation frequency. Evaluating PSC amplitude can be done either by deconvolving EPSP or simply by measuring the amplitudes of EPSP onsets from the trough just before to the peak, which is a good proxy of PSC amplitude.

2) The manuscript would greatly benefit from an additional silencing experiment that directly tests to what extent the descending inputs indeed influence the onset and timing of IC shell neuron activity in vivo. One possibility would be to silence layer 5 AC neurons by expressing halo-rhodopsin. Another possibility is cooling. Alternatively, the claims about the functional role of centrifugal inputs should be toned down, though without this key experiment the impact of the paper would then be reduced.

3) Most figures with current clamp experiments: it is unclear why no action potentials were elicited while stimulating the ascending and descending excitatory inputs, as the electrode solution did not contain QX314 for current clamp experiments. In particular, the larger EPSPs during repetitive stimulation and activation of NMDA receptors could be enhanced by Na+-channel activation. Injecting hyperpolarizing currents may help, but does not prevent Na+-channel activation.

4) Figure 2: the method to estimate the number of input fibres is classically used with electrical stimulation of inputs. It has not been confirmed whether with this stimulation paradigm higher LED intensity results in activation of more fibres. Alternatively, increasing LED intensities may elicit several action potentials in the same fibre. This is rather difficult to differentiate in the current-clamp recording condition as integration times are rather long, leading to halfwidth of EPSP in the range of 20-50 ms. Doing the same recordings in the voltage-clamp condition, which elicits much shorter EPSCs, would be much more convincing. Alternatively, please demonstrate that EPSC kinetics are similar to those observed for single spikes.

5) R-CPP, is rather selective for NMDA receptors containing the GluN2 subunit. Using a less selective NMDA receptor antagonist, that also blocks GluN1 subunits, may reveal more NMDA receptors being activated by the AC projection. This should be tested.

6) The discussion does not refer to the study by Xiong XR, Liang F, Zingg B, Ji X, Ibrahim LA, Tao HW, Zhang LI (2015) Auditory cortex controls sound-driven innate defence behaviour through corticofugal projections to inferior colliculus. Nat Comm 6:7224., which may have important implications for the function of this circuit.

7) The first paragraph of the discussion addresses how this descending input may integrate with sound rhythms and internal cortical rhythms. Although this is an important point of the discussion, it is too vague to discuss it in the first paragraph and should probably be combined with the last paragraph of the discussion.

8) The authors claim several times that the descending projection to the central nucleus of the IC is weak and they neglect its possible functional role. It is true that the large AC projection primarily targets the cortical regions or shell of IC, but it is beyond doubt that it also targets the central nucleus (e.g. Saldaña's studies). In the absence of recordings from the central nucleus, which would be challeneging to do, the authors should to tone down these comments throughout the manuscript. Also, the reason for a 'weak' projection to the central nucleus may be the size and location of the injections made in the auditory cortex. Thus, it is necessary to show Chronos injections sites if available.

9) The authors should consider referring to the non-lemniscal IC in the title of the manuscript as most of the data is related to this area.

10) While the dogma is that the descending projections are glutamatergic, the authors may care to read a recently published paper

https://www.frontiersin.org/articles/10.3389/fncir.2021.714780/full

these authors challenge this view where they have found inhibitory long-range VIP-GABAergic neurons that target IC. Please comment about how this projection may have influence the results in the present study?

*Reviewer #1 (Recommendations for the authors):*

1. The authors finish the first paragraph of the results (Figure 1) with this sentence: "These data indicate that descending excitation can be quite potent, such that even brief and presumably sparse activity of auditory cortico-collicular neurons could in principle drive IC efferent signals independently of ascending inputs."

This is over interpreted at that stage as the authors perform a massive stimulation of AC. Realistic estimates of single fiber strength reported afterwards will help refine this statement, which should be adjusted accordingly.

2. The description of short term plasticity in figure 4 is too superficial, and not precise enough to be used in models. Synapses seem to be depressing but not too much. Ideally one would fit the Markram Tsodyks model for synaptic depression, but that may be out of scope. The authors should at least extract better the adaptation kinetics of PSC trains, for example by reporting the ratio between first and last PSC (not the PSP) for the two repetition frequencies tested, or measuring a decay time constant for each stimulation frequency. Evaluating PSC amplitude can be done either by deconvolving EPSP or simply by measuring the amplitudes of EPSP onsets from the trough just before to the peak which is a good proxy of PSC amplitude.

*Reviewer #2 (Recommendations for the authors):*

1. The manuscript would greatly benefit from an additional experiment that directly tests to what extent the descending inputs do indeed influence the onset and timing of IC shell neuron activity. One possibility would be to silence layer 5 AC neurons by expressing halo-rhodopsin. This should influence the magnitude and the timing of IC shell neurons during sound presentation.

2. Figure 1B: there is no information on what the green and the yellow structures represent. Is red the GFP-Chronos? If yes, I would argue that there is substantial GFP/Chronos-positive input also in the central IC.

3. Most figures with current clamp experiments: it is unclear to me why no action potentials were elicited while stimulating the ascending and descending excitatory inputs, as the electrode solution did not contain QX314 for current clamp experiments. Especially the larger EPSPs during repetitive stimulation and activation of NMDA receptors could be enhanced by Na+-channel activation. Injecting hyperpolarizing currents may help, but does not prevent Na+-channel activation.

4. Figure 2: the method to estimate the number of input fibres is classically used with electrical stimulation of inputs. Whether with this stimulation paradigm higher LED intensity activates more fibres has not been confirmed. Alternatively, increasing LED intensities may elicit several action potentials in the same fibre. This is rather difficult to differentiate in the current-clamp condition recording condition as integration times are rather long leading to halfwidth of EPSP in the range of 20-50 ms. Doing the same recordings in the voltage-clamp condition, which elicits much shorter EPSCs, would be much more convincing.

5. R-CPP, is rather selective for NMDA receptors containing the GluN2 subunit. Using a less selective NMDA receptor antagonist, that also blocks GluN1 subunits, may reveal more NMDA receptors being activated by the AC projection.

6. Figure 7 and 8: Figure 3 convincingly shows that shell IC neurons with very different integrative membrane properties receive descending projections. Some data on how these this affects integration of inputs would strengthen the main message of the manuscript.

7. The discussion does not refer to the study by Xiong XR, Liang F, Zingg B, Ji X, Ibrahim LA, Tao HW, Zhang LI (2015) Auditory cortex controls sound-driven innate defence behaviour through corticofugal projections to inferior colliculus. Nat Comm 6:7224., which may have important implications for the function of this circuit.

8. The first paragraph of the discussion addresses how this descending input may integrate with sound rhythms and internal cortical rhythms. Although this is an important point of the discussion it is too vague to discuss it in the first paragraph and should probably combined with the last paragraph of the discussion.

*Reviewer #3 (Recommendations for the authors):*

1. The authors may care to cite a recent study by Valdes-Baizabal and colleagues on cell types in the dorsal IC. The study is quite modest but should be includeed for the sake of completeness.

2. The authors tends to have lots of refs and citation to previous works in the Results. While this gives a good perspective to the results, I wonder if they distract the reader a bit and whether they could be moved to the Introduction, so that the Results section is more focused and related to the actual results. I feel the introduction could be improved in these general terms.

---

## [Author Response]

Essential revisions:1) The description of short-term plasticity in figure 4 is too superficial, and not precise enough to be used in models. Synapses seem to be depressing but not too much. Ideally one would fit the Markram Tsodyks model for synaptic depression, but that may be out of scope. The authors should at least extract better the adaptation kinetics of PSC trains, for example by reporting the ratio between first and last PSC (not the PSP) for the two repetition frequencies tested, or by measuring a decay time constant for each stimulation frequency. Evaluating PSC amplitude can be done either by deconvolving EPSP or simply by measuring the amplitudes of EPSP onsets from the trough just before to the peak, which is a good proxy of PSC amplitude.

We thank Reviewer #1 for raising this point regarding applicability to future modeling efforts. We have now performed additional measurements of instantaneous amplitude for each EPSP in the stimulus trains, fit the average data with mono-exponential time constants, and report the fit parameters. Additionally, we plot the ratio of first and last instantaneous EPSP amplitudes in 20 and 50 Hz trains. These data are now presented as Figure 4 —figure supplement 1.

2) The manuscript would greatly benefit from an additional silencing experiment that directly tests to what extent the descending inputs indeed influence the onset and timing of IC shell neuron activity in vivo. One possibility would be to silence layer 5 AC neurons by expressing halo-rhodopsin. Another possibility is cooling. Alternatively, the claims about the functional role of centrifugal inputs should be toned down, though without this key experiment the impact of the paper would then be reduced.

A number of studies over the past several decades have shown that auditory cortex inactivation changes the magnitude, timing, and/or response selectivity of IC neurons (Yan and Suga, 1999; Nwabueze-Ogbo et al., 2002; Popelár et al., 2003; Nakamoto et al., 2008, 2010; Anderson and Malmierca, 2013; Popelář et al., 2016; Weible et al., 2020). Indeed, these classic experiments were a major driving force for our current study of how corticofugal transmission impacts IC neuron excitability. We therefore felt that including a cortical inactivation experiment in the current paper would be largely confirmatory of prior knowledge. We also worried that cortical inactivation, even using cell-type specific manipulations, would have brain-wide effects which preclude unambiguously ascribing the effects to a silencing of auditory cortico-collicular transmission (Li et al., 2019; Andrei et al., 2021; Slonina et al., 2021).

However, we also acknowledge that a loss of function experiment helps place our cellular-level findings in a larger systems-level context, thereby broadening the impact of our work. We have thus performed new experiments to show that silencing the auditory cortex reversibly slows the rising phase of sound-evoked field potentials and delays the onset of multi-unit spiking in the IC of awake mice (Figure 7 —figure supplement 2). Even given the interpretive caveats (which we raise in the Discussion), these results are quite interesting: The data suggest that spontaneous activity in auditory cortex *prior* to sound onset might predictively impact IC neuron responses to ascending inputs. We performed this experiment with cooling instead of halorhodopsin for the following reasons.

– Layer 5 neurons are ~600-800 µm below the cortical surface. Activating halo-rhodopsin this deep in the cortex either requires invasive optic fiber implants that will damage the apical dendrites of corticofugal neurons, or alternatively, high light power with known off-target effects (~10-30 mW; Owen et al., 2019). By contrast, cooling enables rapid and reversible silencing while preserving tissue integrity.

– Although cooling is not specific to L5, a similar issue of non-specificity applies to “layer-specific” manipulations given the translaminar, cross-columnar, and brain-wide projection pattern of L5 axons: In somatosensory cortex, inactivating L5 pyramidal neurons in fact *increases* pyramidal neuron excitability in all cortical layers, including other L5 neurons not expressing the inhibitory opsin. This effect is due to the powerful interactions between L5 pyramidal neurons and fast-spiking interneurons (Vecchia et al., 2020). Similarly, a study published in *eLife* during our manuscript revision elegantly shows how optogenetic inactivation of a small population of macaque V1 neurons causes heterogeneous effects in neighboring, non-inactivated cortical tissue (Andrei et al., 2021). Unfortunately, the extent of cortical connectivity, rather than spread of light in tissue, is the immutable limiting factor for any optogenetic manipulation (Li et al., 2019). Given this background, we would expect Layer 5 silencing to similarly drive auditory cortical hyper-excitability as in Vecchia et al; it would be difficult to unambiguously interpret any effect in the IC as due to a reduction, rather than a potential increase, of cortical activity. Thus, we reasoned that an “old-school” approach such as cortical cooling provides a more interpretable result in this particular context.

3) Most figures with current clamp experiments: it is unclear why no action potentials were elicited while stimulating the ascending and descending excitatory inputs, as the electrode solution did not contain QX314 for current clamp experiments. In particular, the larger EPSPs during repetitive stimulation and activation of NMDA receptors could be enhanced by Na+-channel activation. Injecting hyperpolarizing currents may help, but does not prevent Na+-channel activation.

We apologize for the confusion. The lack of APs during stimulation is due to the fact that stimulation strengths were explicitly titrated to remain sub-threshold on most trials during repetitive stimulation. We now add this information to the Methods (lines 682-684 of the revised manuscript). We acknowledge that we have not formally investigated the possible contribution of other dendritic non-linearities such as voltage-gated Na^+^ or Ca^2+^ channels, and we now mention this limitation in our revised Discussion. However, it is worth noting that our observed non-linearity is largely blocked by the NMDAR antagonist R-CPP, indicating that NMDARs mediate the bulk of our integrative effect.

4) Figure 2: the method to estimate the number of input fibres is classically used with electrical stimulation of inputs. It has not been confirmed whether with this stimulation paradigm higher LED intensity results in activation of more fibres. Alternatively, increasing LED intensities may elicit several action potentials in the same fibre. This is rather difficult to differentiate in the current-clamp recording condition as integration times are rather long, leading to halfwidth of EPSP in the range of 20-50 ms. Doing the same recordings in the voltage-clamp condition, which elicits much shorter EPSCs, would be much more convincing. Alternatively, please demonstrate that EPSC kinetics are similar to those observed for single spikes.

Reviewer #2 raises an important point. If stronger light pulses were to increase the duration of activity in single axons, we would expect the EPSC rise time and half-width to scale with LED intensity. We have performed this critical control experiment and find that the rise-time and half-width of cortical EPSCs are similar whether elicited at threshold or via maximal intensity stimulation (Figure 2 —figure supplement 1).

5) R-CPP, is rather selective for NMDA receptors containing the GluN2 subunit. Using a less selective NMDA receptor antagonist, that also blocks GluN1 subunits, may reveal more NMDA receptors being activated by the AC projection. This should be tested.

This is another good point raised by Reviewer #2: Our interpretation rests upon the assumption that auditory cortico-collicular synapses do not contain R-CPP resistant NMDA receptors.

However, we believe that the proposed experiment, testing the effect of a less specific NMDAR antagonist acting at the GluN1 subunit, would be difficult to interpret: Antagonists of the GluN1 subunit also non-specifically inhibit glutamate release from presynaptic terminals by binding to an unidentified substrate (Huang et al., 2011). This effect, which strikingly persists in GluN1 KO mice, occurs at many glutamatergic synapses throughout the brain. Given this background, any result showing a greater apparent reduction of descending EPSPs by the proposed less-selective antagonist compared to R-CPP would be confounded by the high likelihood of non-specific, presynaptic effects. Nevertheless, this is an important issue which we address below with rhetoric and additional experiments.

R-CPP exhibits nanomolar Ki values at GluN2A, B, and C containing NMDARs; the concentration we apply (5 µM) should saturate these receptors (Feng et al., 2005). The alternative hypothesis is that descending cortical synapses mostly contain GluN1/GluN2D heteromers which are less sensitive to R-CPP (Feng et al., 2005) and have notoriously slow deactivation kinetics (~3 s; Misra et al., 2000). However, the range of decay taus for descending EPSCs at positive potentials is one to two orders of magnitude faster (~50-200 ms; our Figure 5 —figure supplement 1), which is inconsistent with this hypothesis.

Furthermore, If the minor effect of R-CPP on descending EPSPs (Figure 5C-H) is due to the presence of R-CPP insensitive, GluN2D containing NMDARs, we would predict that auditory cortical EPSCs at positive potentials (where the NMDA component dominates due to relief of the Mg^2+^ block) should be similarly insensitive to 5 µM R-CPP. To the contrary, we now show that 5 µM R-CPP profoundly accelerates the time-course of auditory cortical EPSCs at positive potentials (Figure 5 —figure supplement 1C,D). Furthermore, the synaptic current remaining in R-CPP is entirely blocked by 10 µm NBQX, indicating that it is fully mediated by AMPA/kainate receptors. We thus conclude that 5 µM R-CPP fully blocks NMDARs at descending cortical synapses. Altogether, our results are more consistent with a differential contribution of NMDARs to ascending and descending transmission rather than a pathway-specific expression of R-CPP insensitive NMDARs.

Finally, we add a new dataset that further supports the differential contribution of NMDARs at ascending and descending inputs (Figure 6 —figure supplement 1): We find that ascending EPSPs evoked with single shocks (as opposed to train stimuli) are more sensitive to R-CPP than descending EPSPs evoked with single light flashes. These data suggest that the differential block observed in Figure 6 is not restricted to an increased contribution of NMDARs during repetitive activation of ascending synapses.

6) The discussion does not refer to the study by Xiong XR, Liang F, Zingg B, Ji X, Ibrahim LA, Tao HW, Zhang LI (2015) Auditory cortex controls sound-driven innate defence behaviour through corticofugal projections to inferior colliculus. Nat Comm 6:7224., which may have important implications for the function of this circuit.

Xiong et al.,’s circuit mapping results were in fact referenced in the second paragraph of the original Discussion (line 373 of the original manuscript). We now add an additional discussion point regarding the important findings of Xiong et al., in the Discussion paragraph titled “Implications for predictive control of tectal activity, synaptic plasticity, and perceptual learning.”

7) The first paragraph of the discussion addresses how this descending input may integrate with sound rhythms and internal cortical rhythms. Although this is an important point of the discussion, it is too vague to discuss it in the first paragraph and should probably be combined with the last paragraph of the discussion.

The text regarding cortical rhythms is now combined with the last paragraph of the Discussion section.

8) The authors claim several times that the descending projection to the central nucleus of the IC is weak and they neglect its possible functional role. It is true that the large AC projection primarily targets the cortical regions or shell of IC, but it is beyond doubt that it also targets the central nucleus (e.g. Saldaña's studies). In the absence of recordings from the central nucleus, which would be challeneging to do, the authors should to tone down these comments throughout the manuscript. Also, the reason for a 'weak' projection to the central nucleus may be the size and location of the injections made in the auditory cortex. Thus, it is necessary to show Chronos injections sites if available.

As suggested we have toned down claims of the “weak” projection to central IC and provide micrographs of Chronos injection sites (Figure 1 —figure supplement 1). These micrographs are from new experiments, as the quality of the original slides shown in the original Figure 1 had degraded. We agree that whole-cell recordings from the central IC in adult mice are quite challenging due to the extensive myelination. In lieu of such functional circuit mapping data, we include a new transsynaptic tracing experiment to label the somata of presumptive postsynaptic targets of auditory cortex neurons in the IC (Figure 1 —figure supplement 2). The data show that the majority of cortico-recipient IC neurons are located in the shell regions. However, a few central IC neurons are indeed labeled. Future studies will be required to test the extent and potency of this direct auditory cortex->central IC projection.

9) The authors should consider referring to the non-lemniscal IC in the title of the manuscript as most of the data is related to this area.

We have changed the title of the paper to Synaptic Mechanisms of Top-Down Control in the Non-Lemniscal Inferior Colliculus.

10) While the dogma is that the descending projections are glutamatergic, the authors may care to read a recently published paperhttps://www.frontiersin.org/articles/10.3389/fncir.2021.714780/fullthese authors challenge this view where they have found inhibitory long-range VIP-GABAergic neurons that target IC. Please comment about how this projection may have influence the results in the present study?

We thank Reviewer #3 for pointing out this new study which does indeed relate to our work. However, we don’t think direct GABAergic projections contributed much, if at all to our results. Indeed, the experiments of Figure 5A did not reveal any inhibitory postsynaptic potentials following bath application of NBQX as one might expect from direct stimulation of VIP-GABA axons (these experiments were performed without SR95531 in the bath). Rather, it may be that the VIP-GABA synapses have low release probability, transmit mainly via non-synaptic diffusion (e.g., spillover), or may primarily release the neuropeptide VIP which would be difficult to detect via whole-cell patch-clamp electrophysiology. We now address this paper in our revised Discussion.

Reviewer #1 (Recommendations for the authors):1. The authors finish the first paragraph of the results (Figure 1) with this sentence: "These data indicate that descending excitation can be quite potent, such that even brief and presumably sparse activity of auditory cortico-collicular neurons could in principle drive IC efferent signals independently of ascending inputs."This is over interpreted at that stage as the authors perform a massive stimulation of AC. Realistic estimates of single fiber strength reported afterwards will help refine this statement, which should be adjusted accordingly.

We have toned down this statement. The first paragraph now finishes with:

“These data indicate that bulk activation of corticofugal neurons triggers potent EPSPs in shell IC neurons. However, under behaviorally relevant conditions, the extent of synaptic depolarization provided by descending inputs will depend on the rate and synchrony of corticofugal neuron firing.”

2. The description of short term plasticity in figure 4 is too superficial, and not precise enough to be used in models. Synapses seem to be depressing but not too much. Ideally one would fit the Markram Tsodyks model for synaptic depression, but that may be out of scope. The authors should at least extract better the adaptation kinetics of PSC trains, for example by reporting the ratio between first and last PSC (not the PSP) for the two repetition frequencies tested, or measuring a decay time constant for each stimulation frequency. Evaluating PSC amplitude can be done either by deconvolving EPSP or simply by measuring the amplitudes of EPSP onsets from the trough just before to the peak which is a good proxy of PSC amplitude.

We address this issue in our revision with a new analysis. Please see our response to Essential Revision #1.

Reviewer #2 (Recommendations for the authors):1. The manuscript would greatly benefit from an additional experiment that directly tests to what extent the descending inputs do indeed influence the onset and timing of IC shell neuron activity. One possibility would be to silence layer 5 AC neurons by expressing halo-rhodopsin. This should influence the magnitude and the timing of IC shell neurons during sound presentation.

We address this issue in our revision with a new experiment. Please see our response to Essential Revision #2.

2. Figure 1B: there is no information on what the green and the yellow structures represent. Is red the GFP-Chronos? If yes, I would argue that there is substantial GFP/Chronos-positive input also in the central IC.

We apologize for the confusion. We have now changed the data presentation to more clearly show the distribution of cortico-collicular axons in shell and central IC (Figure 1 —figure supplement 1). The data show that most axons are located in the shell IC, although the central IC does contain a few sparsely distributed fibers. We also include a new transsynaptic labeling experiment showing that most postsynaptic targets of auditory cortex reside in shell IC (Figure 1 —figure supplement 2). Consistent with the aforementioned axonal labeling, this experiment shows that a few transsynaptically labeled neurons are indeed observable in central IC.

3. Most figures with current clamp experiments: it is unclear to me why no action potentials were elicited while stimulating the ascending and descending excitatory inputs, as the electrode solution did not contain QX314 for current clamp experiments. Especially the larger EPSPs during repetitive stimulation and activation of NMDA receptors could be enhanced by Na+-channel activation. Injecting hyperpolarizing currents may help, but does not prevent Na+-channel activation.

We clarify this confusion in our revision and add a new Discussion point. Please see our response to Essential Revision #3.

4. Figure 2: the method to estimate the number of input fibres is classically used with electrical stimulation of inputs. Whether with this stimulation paradigm higher LED intensity activates more fibres has not been confirmed. Alternatively, increasing LED intensities may elicit several action potentials in the same fibre. This is rather difficult to differentiate in the current-clamp condition recording condition as integration times are rather long leading to halfwidth of EPSP in the range of 20-50 ms. Doing the same recordings in the voltage-clamp condition, which elicits much shorter EPSCs, would be much more convincing.

We address this issue in our revision with a new experiment. Please see our response to Essential Revision #4.

5. R-CPP, is rather selective for NMDA receptors containing the GluN2 subunit. Using a less selective NMDA receptor antagonist, that also blocks GluN1 subunits, may reveal more NMDA receptors being activated by the AC projection.

We address this concern in our revision with a new experiment. Please see our response to Essential Revision #5.

6. Figure 7 and 8: Figure 3 convincingly shows that shell IC neurons with very different integrative membrane properties receive descending projections. Some data on how these this affects integration of inputs would strengthen the main message of the manuscript.

We agree that correlating intrinsic and synaptic properties could reveal something interesting. However, our initial analyses (Figure 3) did not show any striking correlation between membrane biophysics and the half-width or amplitude of descending EPSPs. As such, we had no a priori basis to hypothesize that synaptic integration differs systematically with measurable membrane properties, and the low-throughput of dual pathway stimulation experiments (Figures 7 and 8) precluded collecting a large dataset needed to convincingly determine if synaptic non-linearities do or do not meaningfully correlate with the cellular biophysics. We acknowledge this as a limitation of our study in our revised Discussion. Future studies, perhaps leveraging cell-type specific markers for different IC neurons (Goyer et al., 2019; Naumov et al., 2019; Silveira et al., 2020; Kreeger et al., 2021), will be required to clarify this issue.

7. The discussion does not refer to the study by Xiong XR, Liang F, Zingg B, Ji X, Ibrahim LA, Tao HW, Zhang LI (2015) Auditory cortex controls sound-driven innate defence behaviour through corticofugal projections to inferior colliculus. Nat Comm 6:7224., which may have important implications for the function of this circuit.

We have appropriately revised our Discussion section. Please see our response to Essential Revision #6.

8. The first paragraph of the discussion addresses how this descending input may integrate with sound rhythms and internal cortical rhythms. Although this is an important point of the discussion it is too vague to discuss it in the first paragraph and should probably combined with the last paragraph of the discussion.

The Discussion section has been revised as suggested. Please see our response to Essential Revision #7.

Reviewer #3 (Recommendations for the authors):1. The authors may care to cite a recent study by Valdes-Baizabal and colleagues on cell types in the dorsal IC. The study is quite modest but should be includeed for the sake of completeness.

We agree, the Valdes-Baizabal paper is quite nice. We now reference this paper with respect to their findings regarding sound-evoked spiking and synaptic latencies in shell IC neurons.

2. The authors tends to have lots of refs and citation to previous works in the Results. While this gives a good perspective to the results, I wonder if they distract the reader a bit and whether they could be moved to the Introduction, so that the Results section is more focused and related to the actual results. I feel the introduction could be improved in these general terms.

As suggested, we have made an effort to remove some superfluous citations in the Results section.

References

Anderson LA, Malmierca MS (2013) The effect of auditory cortex deactivation on stimulus-specific adaptation in the inferior colliculus of the rat. Eur J Neurosci 37:52–62.

Andrei AR, Debes S, Chelaru M, Liu X, Rodarte E, Spudich JL, Janz R, Dragoi V (2021) Heterogeneous side effects of cortical inactivation in behaving animals. *eLife* 10:e66400.

Feng B, Morley RM, Jane DE, Monaghan DT (2005) The effect of competitive antagonist chain length on NMDA receptor subunit selectivity. Neuropharmacology 48:354–359.

Goyer D, Silveira MA, George AP, Beebe NL, Edelbrock RM, Malinski PT, Schofield BR, Roberts MT (2019) A novel class of inferior colliculus principal neurons labeled in vasoactive intestinal peptide-Cre mice. *eLife* 8:e43770.

Huang YH, Ishikawa M, Lee BR, Nakanishi N, Schlüter OM, Dong Y (2011) Searching for presynaptic NMDA receptors in the nucleus accumbens. J Neurosci Off J Soc Neurosci 31:18453–18463.

Kreeger LJ, Connelly CJ, Mehta P, Zemelman BV, Golding NL (2021) Excitatory cholecystokinin neurons of the midbrain integrate diverse temporal responses and drive auditory thalamic subdomains. Proc Natl Acad Sci U S A 118:e2007724118.

Li N, Chen S, Guo ZV, Chen H, Huo Y, Inagaki HK, Chen G, Davis C, Hansel D, Guo C, Svoboda K (2019) Spatiotemporal constraints on optogenetic inactivation in cortical circuits. *eLife* 8:e48622.

Misra C, Brickley SG, Wyllie DJ, Cull-Candy SG (2000) Slow deactivation kinetics of NMDA receptors containing NR1 and NR2D subunits in rat cerebellar Purkinje cells. J Physiol 525 Pt 2:299–305.

Nakamoto KT, Jones SJ, Palmer AR (2008) Descending projections from auditory cortex modulate sensitivity in the midbrain to cues for spatial position. J Neurophysiol 99:2347–2356.

Nakamoto KT, Shackleton TM, Palmer AR (2010) Responses in the inferior colliculus of the guinea pig to concurrent harmonic series and the effect of inactivation of descending controls. J Neurophysiol 103:2050–2061.

Naumov V, Heyd J, de Arnal F, Koch U (2019) Analysis of excitatory and inhibitory neuron types in the inferior colliculus based on Ih properties. J Neurophysiol 121:2126–2139.

Nwabueze-Ogbo FC, Popelár J, Syka J (2002) Changes in the acoustically evoked activity in the inferior colliculus of the rat after functional ablation of the auditory cortex. Physiol Res 51 Suppl 1:S95–S104.

Owen SF, Liu MH, Kreitzer AC (2019) Thermal constraints on in vivo optogenetic manipulations. Nat Neurosci 22:1061–1065.

Popelár J, Nwabueze-Ogbo FC, Syka J (2003) Changes in neuronal activity of the inferior colliculus in rat after temporal inactivation of the auditory cortex. Physiol Res 52:615–628.

Popelář J, Šuta D, Lindovský J, Bureš Z, Pysanenko K, Chumak T, Syka J (2016) Cooling of the auditory cortex modifies neuronal activity in the inferior colliculus in rats. Hear Res 332:7–16.

Silveira MA, Anair JD, Beebe NL, Mirjalili P, Schofield BR, Roberts MT (2020) Neuropeptide Y Expression Defines a Novel Class of GABAergic Projection Neuron in the Inferior Colliculus. J Neurosci 40:4685–4699.

Slonina ZA, Poole KC, Bizley JK (2021) What can we learn from inactivation studies? Lessons from auditory cortex. Trends Neurosci:S0166-2236(21)00203-4.

Vecchia D, Beltramo R, Vallone F, Chéreau R, Forli A, Molano-Mazón M, Bawa T, Binini N, Moretti C, Holtmaat A, Panzeri S, Fellin T (2020) Temporal Sharpening of Sensory Responses by Layer V in the Mouse Primary Somatosensory Cortex. Curr Biol CB 30:1589-1599.e10.

Weible AP, Yavorska I, Wehr M (2020) A Cortico-Collicular Amplification Mechanism for Gap Detection. Cereb Cortex N Y N 1991 30:3590–3607.

Yan J, Suga N (1999) Corticofugal Amplification of Facilitative Auditory Responses of Subcortical Combination-Sensitive Neurons in the Mustached Bat. J Neurophysiol 81:817–824.